# communications
# engineering

# Projective diffeomorphic mapping of molecular digital pathology with tissue MRI

Kaitlin M. Stouffer [1✉], Menno P. Witter [2], Daniel J. Tward [3,4] & Michael I. Miller [1,4]

Reconstructing dense 3D anatomical coordinates from 2D projective measurements has become a central problem in digital pathology for both animal models and human studies. Here we describe Projective Large Deformation Diffeomorphic Metric Mapping (LDDMM), a technique which projects diffeomorphic mappings of dense human magnetic resonance imaging (MRI) atlases at tissue scales onto sparse measurements at micrometre scales associated with histological and more general optical imaging modalities. We solve the problem of dense mapping surjectively onto histological sections by incorporating technologies for crossing modalities that use nonlinear scattering transforms to represent multiple radiomic-like textures at micron scales, together with a Gaussian mixture-model framework for modeling tears and distortions associated to each section. We highlight the significance of our method through incorporation of neuropathological measures and MRI, of relevance to the development of biomarkers for Alzheimer's disease and one instance of the integration of imaging data across the scales of clinical imaging and digital pathology.

[1] Department of Biomedical Engineering, Johns Hopkins University, Baltimore, MD, USA. [2] Kavli Institute for Systems Neuroscience, Norwegian University of Science and Technology, Trondheim, Torgarden, Norway. [3] Departments of Computational Medicine and Neurology, University of California, Los Angeles, CA, USA. [4] These authors contributed equally: Daniel J. Tward, Michael I. Miller. ✉email: kstouff4@jhmi.edu

The past decade has ushered an omics revolution into bio-medical research, with high yields of data ranging from microscopic to macroscopic scales. Modern machine learning methods coupled with image processing have enabled the integration of pathomics data extracted from digital pathology technologies with radiomics data extracted from lower resolution imaging technologies such as magnetic resonance imaging (MRI) in a number of niche applications, such as those within the domain of cancer diagnostics and prognostics[1]. However, approaches remain widely varied across applications and often require particularities in image acquisition and image type, such as block face imaging[2], to facilitate alignment between imaging modalities[3]. The obstacle of registering spatially incomplete sets of 2D (or 3D) images to a dense 3D atlas remains a particular challenge in both animal and human settings, with no approach yet amenable to synthesizing the imaging data from the spectrum of technologies within this domain.

The first major contribution of this work is the introduction of a new class of image-based diffeomorphometry methods which we term Projective Large Deformation Diffeomorphic Metric Mapping (Projective LDDMM) for aligning sparse sets of image captures to 3D coordinate systems across micron and millimeter scales. We focus on the registration of 3D MRI with 2D digital histology, as representative of the class of multi-scale, multi-modality mapping in biomedical research including traditional light microscopy mapping to dense reference atlases[4–7], light sheet methods[8], deep tissue imaging[9], and spatial transcriptomics[10]. We formulate the mapping of dense atlases to sparse images problem using the random orbit model of computational anatomy[11–13] in which the space of dense anatomies $I \in \mathcal{I}$ is modeled as an orbit of a 3D template under the group of diffeomorphisms. Projective LDDMM models the sparse 2D observables not as an element of the orbit $\mathcal{I}$ but rather a random deformation in dense 3D coordinates composed with a mea-surement projection to sparse coordinates. This implies that the random orbit model encompasses the composition of two observation channels: one for projection and one for post-projection processing, such as the steps involved with histological staining and slide preparation. While LDDMM provides the geodesic metric[14,15] on the orbit of 3D anatomies, there is no symmetry between the observable and the template, in general, and there should not be. This departs substantially from sym-metric methods.

The second major contribution of this work is the introduction of a photometric transformation between modalities via a Scat-tering Transform that simultaneously achieves correspondence between contrasts at different scales, analogous to those captured by convolutional neural networks (CNNs), but without the need for training data or expensive computational resources[16]. For alignment specifically of modes of histology to MRI, cross-modality similarity modeling is essential. Several strategies for representing image similarity have emerged including cross-correlation[17] and local textural characteristics[18]. We previously used a polynomial transformation to accommodate crossing modalities at a single (low) resolution[19]. Others have used a variety of machine learning approaches to cross contrasts both in 2D and 3D, again between images at the same scale[20–22]. Here, to accommodate crossing contrast modalities at different scales, particularly in the context of limited training data, we expand histology image space to a span of non-linear discriminative fil-tered images via Mallat's Scattering Transform[16]. These filtered images are efficiently computed directly from the sample histol-ogy images. They are at the lower resolution of MRI, but repre-sent local, and specifically non-linear, radiomic textures at histological scales, unlike typical linearly downsampled images, and thus, can effectively be used to predict MRI contrast.

As histological images carry large numbers of imperfections with tears, image stitching, and lighting variations, we addition-ally estimate high-dimensional geometric transformations in the image plane through the introduction of Gaussian mixture models. As in previous work[19], these Gaussian mixtures models interpret image locations in each histological slice as matching tissue, background, or artifact. Here, we depart from this and other previous work[4,19] in estimating high-dimensional diffeo-morphisms in the image plane rather than rigid motions to encapsulate the distortion that occurs following tissue sectioning in histological processing. We proceed by way of the Expectation-Maximization (EM) algorithm[23] in estimating deformations that prioritize image matching at locations that are, in turn, estimated more likely to be matching tissue.

Here, we use Projective LDDMM to reconstruct the 3D geo-metry of 2D histological sections taken from the medial temporal lobe (MTL) of a brain with advanced Alzheimer's disease (AD). Efforts to diagnose and manage AD earlier in its disease course have centered on the identification of biomarkers[24]—measures shown to correlate with disease course, but without establishment in AD's pathological underpinnings of misfolded proteins (tau tangles and amyloid-beta (Aβ) plaques)[25–27]. As one application of our method, the reconstruction of a complete spatial profile of tau pathology at the micron level is necessary for validating such biomarkers as entorhinal cortex (ERC) thinning[28]. Following registration, we extract quantities of tau pathology using a machine learning-based approach that are mapped to 3D via the correspondences yielded by Projective LDDMM. We demonstrate the efficacy of modeling these quantities using a measure-based framework[29] as befits resampling the quantities at different scales and within both 2D manifolds and 3D volumes. We additionally smoothe densities over the surface of MTL regions by expanding them in a respective basis for each surface, generated via the Laplace-Beltrami operator, which moves from the classical Euclidean sines and cosines to complete orthonormal bases for smooth curvilinear manifolds, such as these surfaces. Together, these tactics facilitate correlation with biomarkers of interest.

## Results

**Projective LDDMM.** In the random orbit model of computa-tional anatomy[11], the unobserved space of human anatomical images, $I : R^3 \to R^r$, is modeled as an orbit under diffeomorphisms of a template

$$I \in \mathcal{I} := \{\varphi \cdot I_{\text{temp}}, \varphi \in G_{\text{diff}}\}, \tag{1}$$

$G_{\text{diff}}$ the group of diffeomorphisms $\varphi : R^3 \to R^3$, which act on images as $\varphi \cdot I = I \circ \varphi^{-1}$. The observables $J : R^3 \to R^q$ are modeled as a random field with mean due to the randomness of diffeo-morphic deformation and measurement process. For different problems of interest, the atlas image is $R^r$-valued with, for instance, $r = 1$ corresponding to single contrast MRI or $r = 6$ for diffusion tensor images (DTI), as mapped with LDDMM in prior work[30,31]. Likewise, observables are $R^q$-valued with $q = 3$ for traditional histological stains corresponding to the red, green, and blue channels, or $q >> 6$ for alternative representations, such as that given by the Scattering Transform[16] encoding meso-scale radiomic textures in histology images (see the "Linear Prediction Algorithm for Crossing Modalities with Scattering Transform" subsection in the Methods). In general, the range space of 3D templates versus targets do not have the same dimension, so $q \neq r$.

Projective LDDMM is characterized by the fact that the observable is not dense in the 3D metric of the brain. Rather, the observable(s) result from either optical or physical sectioning, as in histological slice preparation, taking LDDMM into the projective setting akin to classical tomography[32]. This sectioning

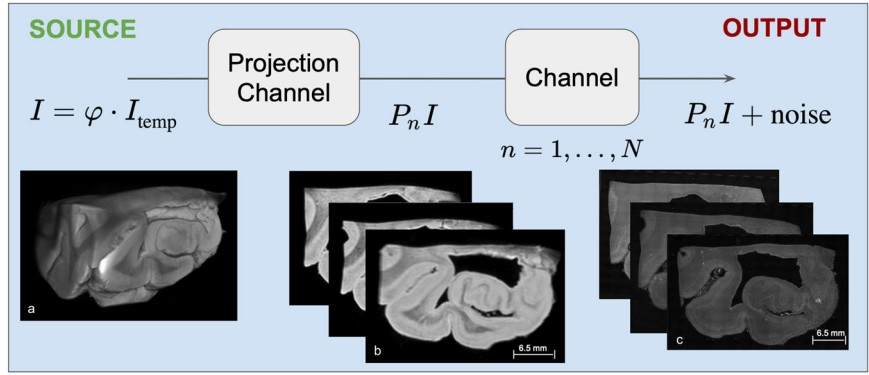

**Fig. 1 The random orbit model extended with projection operator. a** 3D MRI template ($I_{\text{temp}}$), **b** idealized projection slices of deformed MRI template, and **c** set of noisy observables $P_n I$ + noise, $n = 1, ..., N$, representative of histological processing of tissue post slicing. Estimated MR contrast of digital histology image shown with post-sectioning parameters capturing observed changes in sulcal width and ventricle shape in histological sections compared with idealized MRI projections of **b**.

lends itself to extending the random orbit model of computational anatomy involving a single channel of Shannon theory to include a second channel, which represents the projective geometry. As shown in Fig. 1, this projection channel precedes a second channel associated with the parameters of processing post sectioning, such as rigid motions and diffeomorphic deformations.

The sample measured observables $J_n(\cdot), n = 1, 2, \ldots$ are a series of (transformed) projections $P_n$ of $I(\cdot)$ on the source space $X \subset R^3$ to measurement space $Y \subset R^d$ (with $d = 1, 2, 3$) defined through the class of point-spread functions associating source to target:

$$P_n I(y) := \int_X p_n(y, dx) I(x), y \in Y, \quad (2a)$$

$$\text{with } J_n = P_n I + \text{noise.} \quad (2b)$$

We generalize the notion of projection to accommodate point-spreads mapping $R^3$ to $R^3$, as in current light sheet methods, and we adopt measure theoretic notation, $p_n(y, dx)$ for describing point-spreads to accommodate those taking the form of generalized functions, such as the delta Dirac. Density notation $\delta(x - x_0) dx$ corresponds to the measure notation $\delta_{x_0}(dx)$, with both yielding $f(x_0)$ when evaluated against a test function $f(x) \in C^0$.

The diffeomorphism $\varphi$ is generated as the solution to the flow

$$\dot{\varphi}_t = v_t(\varphi_t), \ \varphi_0 = \text{Id}, \quad (3)$$

where Id is the identity map and with velocity field $v_t, t \in [0, 1]$ controlling the flow constrained to be an element of a smooth reproducing kernel Hilbert space $(V, \|\cdot\|_V^2)$ with the entire path square integrable $\int_0^1 \|v_t\|_V^2 dt < \infty$ ensuring smoothness and existence of the inverse[33].

This gives us the first variational problem of Projective LDDMM, with $\|\cdot\|_2$ defined to be the L2 norm.

Variational problem 1: (Projective LDDMM)

$$\dot{\varphi}_t = v_t \circ \varphi_t, \ \varphi_0 = \text{Id}, \ I = I_{\text{temp}} \circ \varphi_1^{-1} \quad (4a)$$

$$P_n : I \mapsto P_n I(y) = \int_X p_n(y, dx) I(x), \ n = 1, \ldots, N \quad (4b)$$

$$\inf_{(v_t)_{0 \le t \le 1} \in L^2([0,1], V)} \int_0^1 \|v_t\|_V^2 dt + \sum_{n=1}^N \|J_n - P_n I\|_2^2 \quad (4c)$$

The model specific to our histology images projects the volume $I(\cdot)$, defined as a function with domain $X \subset R^3$, to parallel sections

$J_n(\cdot)$ on $Y \subset R^2$, along the third ($z$) dimension, with coordinates, $z_n, n = 1, \ldots, N$. This represents a surjection from source space $X$ to target space $Y$, where regions of the 3D volume remain unmapped to the set of target sections. For this, we define Dirac point-spreads from $\delta_x$ applying to infinitesimal volumes in space ($dx$) with $\delta_x(dx)$ equal to 1 if $x \in dx$, and 0, otherwise. Our Dirac point-spreads $p_n(y, dx) = \delta_{y, z_n}(dx)$ with $(y, z_n) = (y^{(1)}, y^{(2)}, z_n) \in R^3$ concentrate on the planes:

$$P_n : I \mapsto P_n I(y) = \int_X \delta_{y, z_n}(dx) I(x), \ y \in Y \subset R^2, \ n = 1, 2, \ldots. \quad (5)$$

In the case that source and target space are of the same dimensionality and our projection operator $P_n$ is the identity, (4c) reduces to the classic variational problem associated to the random orbit model[11]. In the following section, we add the complexity of expanding images in a basis for crossing modalities of histology and MRI. While we choose to expand our target images, a similar expansion of the template image yields a span of possible templates, achieving the setting of multi-atlas models popularly used[34].

**Scattering transforms for digital pathology.** In digital pathology, different imaging contrasts emerge from the variety of stains used to elucidate different molecules, such as myelin (LFB), RNA (Nissl), amyloid (6E10), and tau (PHF-1). Consequently, comparable representation of one stain to another and to images of other modalities requires crossing contrasts and scales of information. To achieve this, we harness Mallat's Scattering Transform to downsample histology images specifically nonlinearly, thereby capturing textural information at high-resolution scales. The Scattering Transform has been shown to be equivalent in structure to CNNs with a particular architecture[16], but without additional training data or extensive time and resources needed to be computed. As a result, it has shown predictive power on par with state-of-the-art deep learning methods in a range of image recognition tasks[35].

Here, for crossing from the range space of a given histology contrast to MRI, we define a predictive basis, $(\psi_n^1(\cdot), \psi_n^2(\cdot), \ldots, 1(\cdot))$, using PCA on a set of non-linear basis images generated from our observables, $J_n(\cdot), n = 1, \ldots, N$ via the Scattering Transform. We expand each histology image to an initial basis of filtered images, each exhibiting a different scale and texture determined by the Scattering Transform[16]. The Scattering Transform mimics the architecture of CNNs by propagating sample images through a series of alternating wavelet convolutions and non-linear modulus operators across scales[16] (see

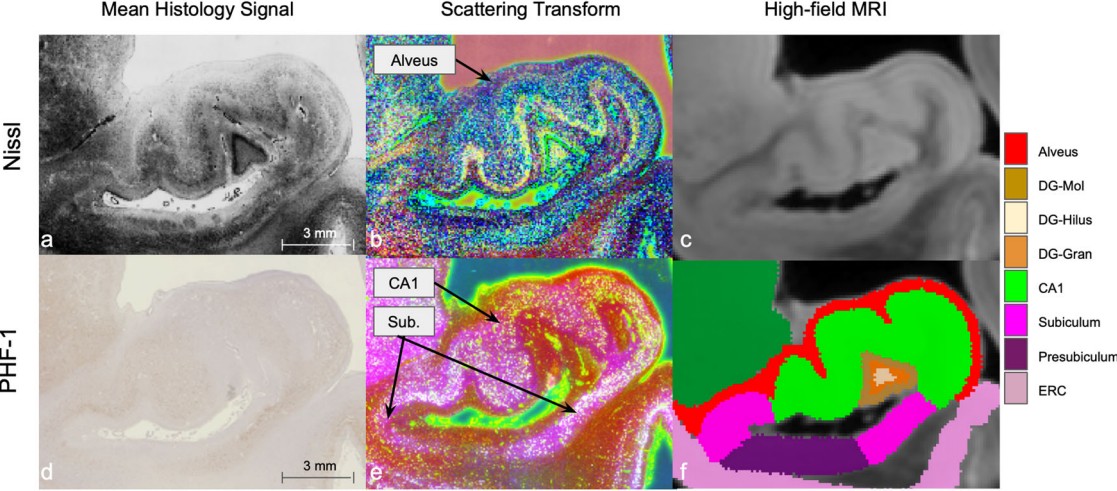

**Fig. 2 Regions discriminated with the scattering transform. a, d** Average signal resulting from linear downsampling of Nissl and PHF-1-stained histology images at 0.002 mm resolution to 0.067 mm resolution. **b, e** Scattering images at 0.067 mm resolution projected onto 3 PCA dimensions and mapped to red, green, blue (RGB). Specific compartments highlighted where scattering representation shows distinguishable boundaries not in average signal. **c, f** High-field MRI slice and manual segmentations showing corresponding contrast and subregion delineations of anatomy in first two columns. Regions shown include those of alveus, dentate gyrus (DG): molecular layer (DG-mol), granular layer (DG-gran), and hilus (DG-hilus), CA1, subiculum, presubiculum, and entorhinal cortex (ERC).

Supplementary Note 3). We then reduce this basis to a discriminating subspace using PCA:

$$\mathcal{S}: J_n(\cdot) \mapsto \mathcal{S}(J_n) = \left( S^1_{J_n}(\cdot), S^2_{J_n}(\cdot), \dots \right) \quad (6a)$$

$$\mathcal{P}: \mathcal{S}(J_n) \mapsto (\psi^1_n, \psi^2_n, \dots, \psi^m_n). \quad (6b)$$

From the non-linear filtered images of (6a), we generate the predictive functions, $\{\psi^j_n\}_{j\in\{1,\dots,m\}}$, using PCA and adding a constant image, $\psi^0_n(\cdot) = 1(\cdot)$. From this basis, we then predict the MRI contrast of our transformed and projected observable as a least-squares minimization, written as $J^\alpha_n(\cdot) := \sum_{k=0}^m \alpha^k_n \psi^k_n(\cdot)$. This process is shown in Supplementary Fig. S1.

Through its non-linear downsampling, the (subsampled) Scattering Transform has the power to distinguish compartments from one another that appear with single contrast in the linearly downsampled image. Examples of this discrimination are shown in Fig. 2 both for Nissl and PHF-1 stained histology images. In the average Nissl image, for instance, the white matter alveus has similar contrast to the neighboring hippocampus and background space, preventing these separate structures from being mapped to different corresponding contrasts in other modalities, such as MRI. Likewise, the separate regions in the hippocampal formation (dentate gyrus and CA fields) appear with similar contrast in the average PHF-1 image, masking the boundary between them, whereas the scattering image clearly manifests this boundary.

For digital pathology and other surjective measurements, there are additional parameters associated with deformation of each tissue section geometry independently in each imaging plane. We denote the associated rigid and/or affine motion in each imaging plane, $\phi_n \in \Phi: R^2 \to R^2$. Estimation of $\alpha_n$ and $\phi_n$ together with $\varphi$ gives Variational Problem 2.

Variational problem 2:

$$\dot{\varphi}_t = v_t \circ \varphi_t, \ \varphi_0 = \mathrm{Id}, \ I = I_{\mathrm{temp}} \circ \varphi_1^{-1} \quad (7a)$$

$$P_n: I \mapsto I(\cdot, z_n), \ n = 1, \dots, N \quad (7b)$$

$$J^\alpha_n(\cdot) := \sum_{k=0}^m \alpha^k_n \psi^k_n(\cdot) \quad (7c)$$

$$\inf_{\substack{(v_t)_{0 \le t \le 1} \in L^2([0,1],V), \\ \alpha_n \in R^m, \phi_n \in \Phi, n=1,\dots,N}} \int_0^1 \|v_t\|^2_V dt + \sum_{n=1}^N \left\| J^\alpha_n - \phi_n \cdot P_n I \right\|^2_2 \quad (7d)$$

For rigid and/or affine motions, as used in modeling block sectioning[4], we apply $\phi_n$ to the histology images and estimate these deformations alternately with deformations of the template (see Supplementary Note 6, Algorithm 1). Herein, we expand our dimensions to non-rigid deformations $\phi_n \in \Phi$ the group of diffeomorphisms on $R^2$. A penalty term is introduced $\int_0^1 \|u_{n,t}\|^2_U dt$ to (7d), with $\dot{\phi}_{n,t} = u_{n,t} \circ \phi_{n,t}, \phi_{n,0} = \mathrm{Id}$. For estimating the cross-modality dimensions $\alpha_n$, we treat them in a maximum-likelihood setting, optimizing them with initial conditions of deformation and image plane dimensions fixed, then solve the variational problem over all of the other dimensions with the $\alpha_n$ estimates fixed (see the "Linear Prediction Algorithm for Crossing Modalities with Scattering Transform" subsection in the Methods). This avoids collapse of the variational problem in these high-dimensional settings.

The independent processing of sections requires the introduction of additional models for interpreting the measurements in each histology tissue section modeling the tissue as foreground, artifacts of tears and distortion, and background (see Tward et al.[19]). The histology is modeled as a conditionally Gaussian mixture-model with means $J^\alpha_n, \mu_A, \mu_B$ representing foreground tissue, artifacts, and background:

$$\mu^n_k(y) = \begin{cases} J^\alpha_n(y) & \text{if } k = 1 \\ \mu_A & \text{if } k = 2 \ , \\ \mu_B & \text{if } k = 3 \end{cases} \quad (8a)$$

with norm-square term in (7d) replaced by

$$\sum_{k=1}^3 \frac{1}{2\sigma^2_k} \left\| (\pi_{n,k})^{\frac{1}{2}} \left( \mu^n_k - \phi_n \cdot P_n I \right) \right\|^2_2. \quad (8b)$$

Weighted least-squares interprets the images weighing each model $\pi_{n,k}(\cdot)$ with $\sum_{k=1}^{3} \pi_k = 1$ and $I = I_{temp} \circ \varphi_1^{-1}$ as above. The weights are estimated iteratively, arising from the E-step of an EM algorithm[23] selecting at each point in the image the appropriate model for giving the spatial field of weights. This iteration corresponds to a generalized EM (GEM) algorithm[23] (see the "Projective LDDMM Algorithm with Multiple Models" subsection in the Methods for proof). The results highlighted in the "Integration of Tau Imaging Data into Multi-scale 3D Maps" subsection in the Results were generated following the approach of this section.

**Optical sectioning**. Additional modes of imaging introduce settings of ideal and non-ideal planar and linear projections fitting the framework for Projective LDDMM. Confocal optical sectioning reconstructs volumes $X \subset R^3$ with models that are fundamentally 3D point-spreads $p_n(y, dx), y \in R^3, dx \subset R^3$ with uncertainty supported over the volumes[36,37], and with imaging focused to $n = 1, \ldots, N$ measurement planes with substantial blur out of plane.

Each projection image, $P_nI(\cdot)$ on $R^3$ of (4b), is constructed both from components within the $n$-th plane of focus and the remaining 3D volume. The relative weight of each component is given by the point-spread $p_n(y, dx)$, which is a function of the detection process and of the microscope's optics[38], such as aperture size, wavelength of light, amplitude of wave on aperture, and aperture orientation[37]. In practice, these point-spreads can be determined experimentally by imaging the spread of a point-source bead[36] in a variety of locations or can be modeled, as in Gibson and Lanni, based on Kirchoff's scalar diffraction laws[37]. Similar optical sectioning effects can be seen with emerging technologies in light sheet microscopy[39], where an anisotropic 3D point-spread governs the construction of each projection image.

Treatment of these imaging technologies within the framework of Projective LDDMM unifies the number of approaches previously taken at reconstructing the underlying image but here, within the setting of introducing an anatomically complex prior distribution as constrained to be within the diffeomorphic orbit of a template. In previous reconstruction models, the projective intensity of (4b) is taken either as the mean field of a Poisson process[36] or as the mean and variance fields of a Gaussian approximation to the Poisson field[38].

In addition to both light sheet microscopy and confocal optical sectioning, classical imaging modalities such as positron emission tomography (PET) and classical tomography (CT) lend themselves to representation within the framework of Projective LDDMM. We illustrate these examples in detail in Supplementary Note 1.

**Integration of Tau imaging data into multi-scale 3D maps**. Using Projective LDDMM, we aligned sets of 2D histology images to corresponding high-field 3D MRI of MTL tissue taken postmortem from a brain sample with advanced AD (see the "Specimen Preparation and Imaging" subsection in the Methods for details on specimen preparation). We quantified relevant pathological measures in digital pathology images as counts of neurofibrillary tangles of hyperphosphorylated tau (NFTs) per cross-sectional area of tissue. NFTs were detected using a machine learning-based algorithm (see the "Tau Pathology Detection" subsection in the Methods) trained and tested to achieve maximum accuracy of detection for the single set of histologically stained tissue slices examined here. Per pixel accuracy of identifying tau was evaluated with 10-fold cross-validation, yielding an average AUC of 0.9860 and accuracy of 0.9729 (see the "Tau Pathology Detection" subsection in the Methods for individual

fold metrics), and final counts of NFTs were validated against a reserved set of regions across 10 separate histology images manually annotated for NFTs and totaling 25 million pixels.

Solutions to Variational Problem 2 (7d) yielded geometric reconstructions of histologically stained tissue in 3D. Figure 3 illustrates individual digitized sections on which NFTs were detected and the resulting positions of these slices following transformation via the estimated geometric mappings to 3D. For future comparison between brain samples and studies, coordinated MRI and histology were mapped to the Mai Paxinos Atlas[40], as demonstrated in Fig. 4 with coronal Mai views shown for an example intersecting histological slice.

Alignment accuracy was evaluated by comparison of manual segmentations on histological images to those deformed from 3D MRI via estimated transformations. Figure 5 illustrates four representative comparisons. We quantified accuracy from these comparisons with Dice overlap and 95th percentile Hausdorff distance for MTL subregions of interest (see Supplementary Note 2). The latter measure ranged from 1.2 mm to 1.8 mm across hippocampal subfields.

Counts of detected NFTs, cross-sectional tissue area, and MTL subregion (from MRI deformed to 2D) were computed in the space of each histology slice. NFT densities (counts of NFTs per cross-sectional tissue area) were modeled as discrete particle measures and transported to the 3D space of the Mai atlas via estimated transformations, $\phi_n, \varphi$, as shown in Fig. 6 (see the "Particle Representation of Histological Data" subsection in the Methods). The flexibility afforded by a measure-based framework as in ref. [29] is reflected in the diverse modes of resampling shown in Fig. 7 both within volumes and over the surface of MTL subregions (see details in Supplementary Note 4).

Resampling via Gaussian kernels yields smoothed NFT densities computed within the dense metric (volume) of the brain at approximate resolutions of MRI. In contrast, spatial variations in NFT density within MTL subregions are visualized as smooth functions over the surface (2D manifold) of each corresponding region, generated via the Laplace-Beltrami operator (see the "Surface Smoothing with Laplace-Beltrami Operator" subsection in the Methods).

## Discussion

We believe this is one of the first image models presented that simultaneously accommodates the generation of sparse or partial image captures from a dense image atlas and an expanded image representation for crossing scales and contrasts. The coupling of approaches used here can be characterized as what is called physics-based machine learning[41], with the key technology used for crossing scales without losing high-resolution information, the Scattering Transform, considered among the class of deep learning and other data-driven empirical models, as described in the "Scattering Transforms for Digital Pathology" subsection in the Results, and the projection operator modeling explicitly the physics of imaging technologies used. While methods exclusively of deep learning have gained popularity in excelling over the years in a range of image classification and processing tasks[42], our coupling of empirical modeling with physics-based modeling has been shown to be more accurate and efficient than either type of approach on its own, particularly in settings of limited training data, as is frequently encountered across applications in biomedicine[41].

Indeed, the inherent generality in our formulation lends itself to use in mapping many other types of imaging modalities in the context of diverse clinical problems, which constitute areas of both current and future work. For instance, we have

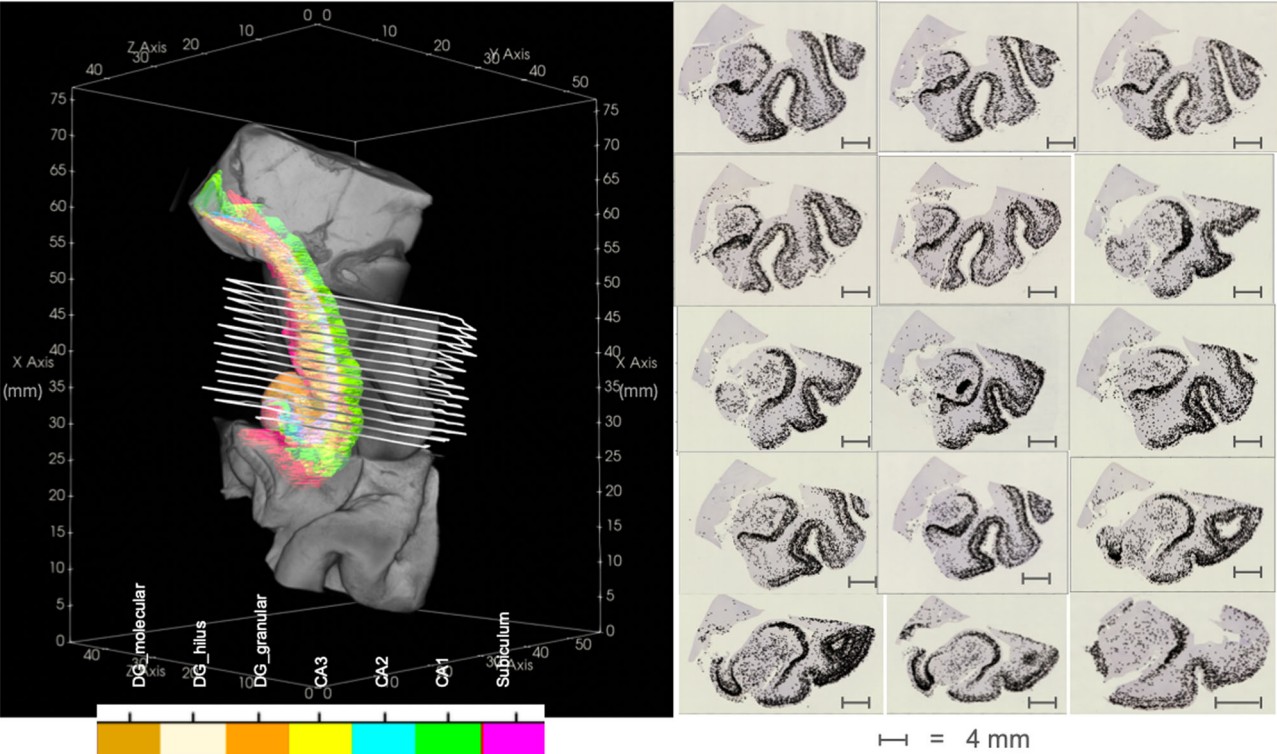

**Fig. 3 Aligned MRI tissue blocks and PHF-1 stained histology sections.** 3D MRI shown with manual segmentations of medial temporal lobe (MTL) subregions including regions of dentate gyrus (DG), hippocampus, and subiculum. Boundary of each histological section on right outlined in white in position following transformation to 3D MRI space. All PHF-1 sections from 1 out of 3 blocks shown. Detected tau tangles plotted as black dots over each histology slice.

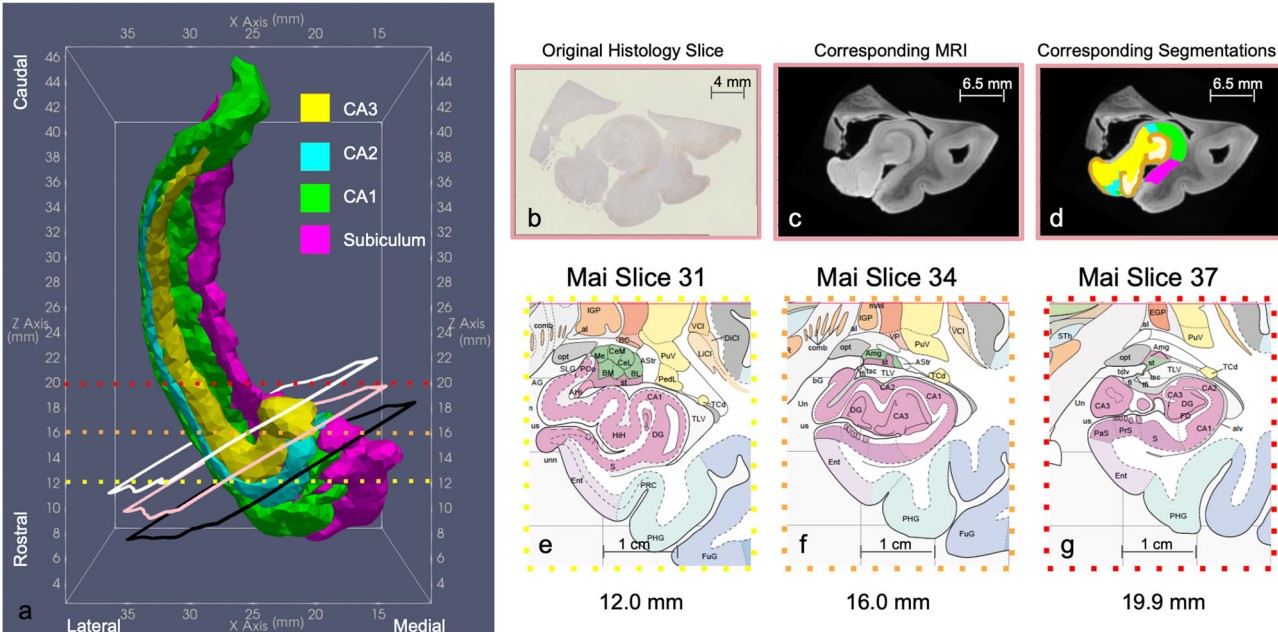

**Fig. 4 3D Geometry of the hippocampus in Mai atlas coordinates. a** 3D Reconstruction of four medial temporal lobe (MTL) subregions for an advanced AD brain in the coordinate space of the Mai Paxinos Atlas. **b** Single section of histology and **c**, **d** corresponding MRI and segmentation projection estimated. **e-g** Intersecting coronal planes taken from the pages of the Mai Atlas[40].

demonstrated the success of this image model at capturing the surjective measures of digital histology images stained with PHF-1 for NFT detection from 3D MRI. The distribution of NFTs across cortical layers and the highlighted cell bodies resulting from the hematoxylin counterstain here may provide an

advantage to other counterstains and pathologies in successfully mapping to MR contrast that similarly distinguishes gray from white matter. However, our successful registration of histological sections from both rostral and central regions of the hippocampus, which harbored different levels of NFT pathology, as

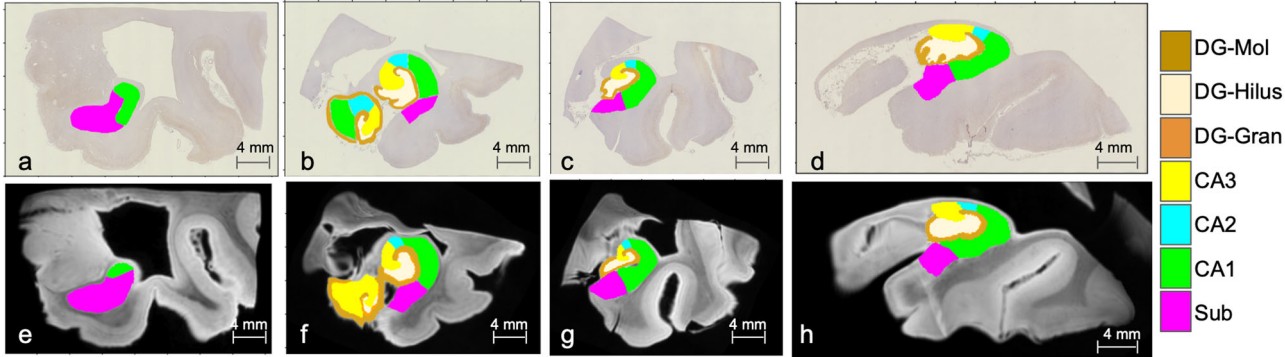

**Fig. 5 Accuracy of alignment between 2D histology slices and 3D MRI. a–d** Selected histology slices with 2D segmentations ordered left to right as rostral to caudal. **e–h** Corresponding MRI slices with 3D segmentations mapped to 2D via transformations $\varphi, \phi_n$. Segmentations of hippocampal formation shown: dentate gyrus (DG) consisting of molecular layer (DG-mol), hilus (DG-hil), and granular layer (DG-gran); subiculum (Sub), and CA1, CA2, CA3 subfields of hippocampus.

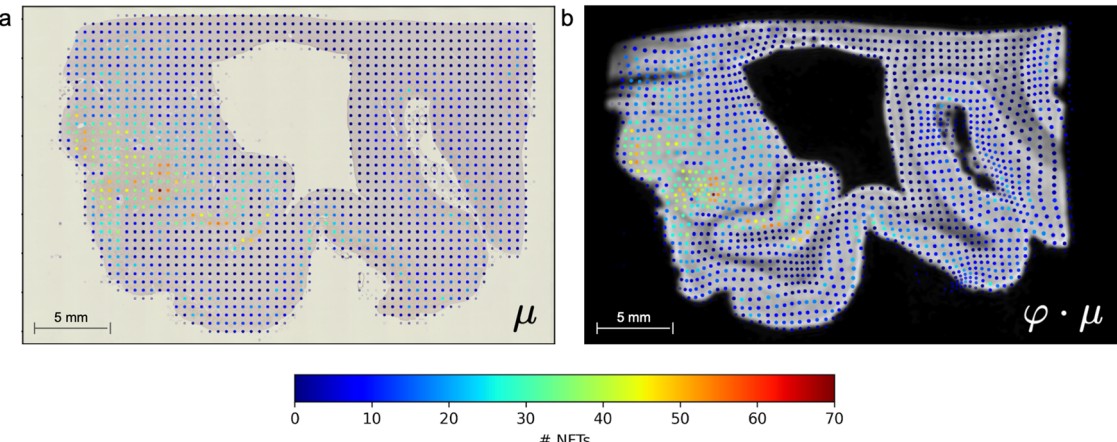

**Fig. 6 Measure-based particle representation before and after diffeomorphic transformation. a** Particles, denoted $\mu$, in native histology space for one section with size of circles proportional to particle weights, defined as the area of tissue sampled, and color of circles reflecting total number of neurofibrillary tau tangles (NFTs) detected in the tissue area for each particle. **b** Particles following transformation to 3D MRI space with weights changing according to varifold action, (14a), and total number of NFTs captured by particle unchanged.

well as the robustness of the Scattering Transform for capturing textures in a range of medical and non-medical images suggest our method could map digital pathology images from other staining protocols and with diverse patterns and presences of pathology to a 3D MR atlas. Furthermore, reconstruction of biological measures in a range of MR atlases, including in vivo T2, DTI, and higher resolution MRI using different RF pulse sequences might afford higher accuracy of alignment to histological images. We are currently mapping 6E10 stained sections highlighting A$\beta$ pathology and hope to examine tauopathies with varying patterns of pathology in the future. Second, as highlighted in the "Optical Sectioning" subsection in the Results and Supplementary Note 1, the projective formulation applies to a range of emerging imaging technologies, such as light sheet methods and those in spatial transcriptomics. Our incorporation of high-dimensional in-plane geometric transformations and Gaussian mixture models for modeling the random effects of tears and tissue processing, occurring independently per section, further accommodates these emerging technologies, which often involve extensive tissue processing. While we estimate smooth deformations between template and target, here, explicit modeling of artifacts and discontinuities through transformations other than diffeomorphisms and with the treatment of the discordant topologies introduced between template and target images represents an additional area for future work.

Clinically, we have appealed to the area of biomarker development in AD[24], by reconstructing 3D maps of NFT density at high resolution, in familiar atlas coordinates, and as smooth distributions over the surface of structures of interest. Reconstruction within the dense 3D metric of the brain, as defined by our high-field MRI atlas, enables pooling of the pathological measures across sets of histology images and resampling within 3D volumes at submillimeter resolutions that exceed that of current technologies in Tau PET. This reconstruction, coupled with the smoothing of densities on the surface of hippocampal subregions by expanding tau measures in a respective basis for each surface, facilitates correlation with many of the reported (smoothed) biomarker measures such as shape and thickness changes[28,43], defined on such surfaces within MR atlases. Similar to Yushkevich et al.[2], we found a strong spatial predominance of tau in the rostral third of the hippocampus. Together with variations in NFT density between hippocampal subregions, this predominance supports Braak's initial observations[25,26] and further underscores the spatial locality of AD. We are currently using our Projective LDDMM technologies to reconstruct tau profiles of an extended set of MTL regions including the ERC and amygdala. By applying our methods to brain samples with intermediate and advanced AD, we plan to examine the specificity of such locality to AD and its temporal changes as compares to published trends in other imaging biomarkers.

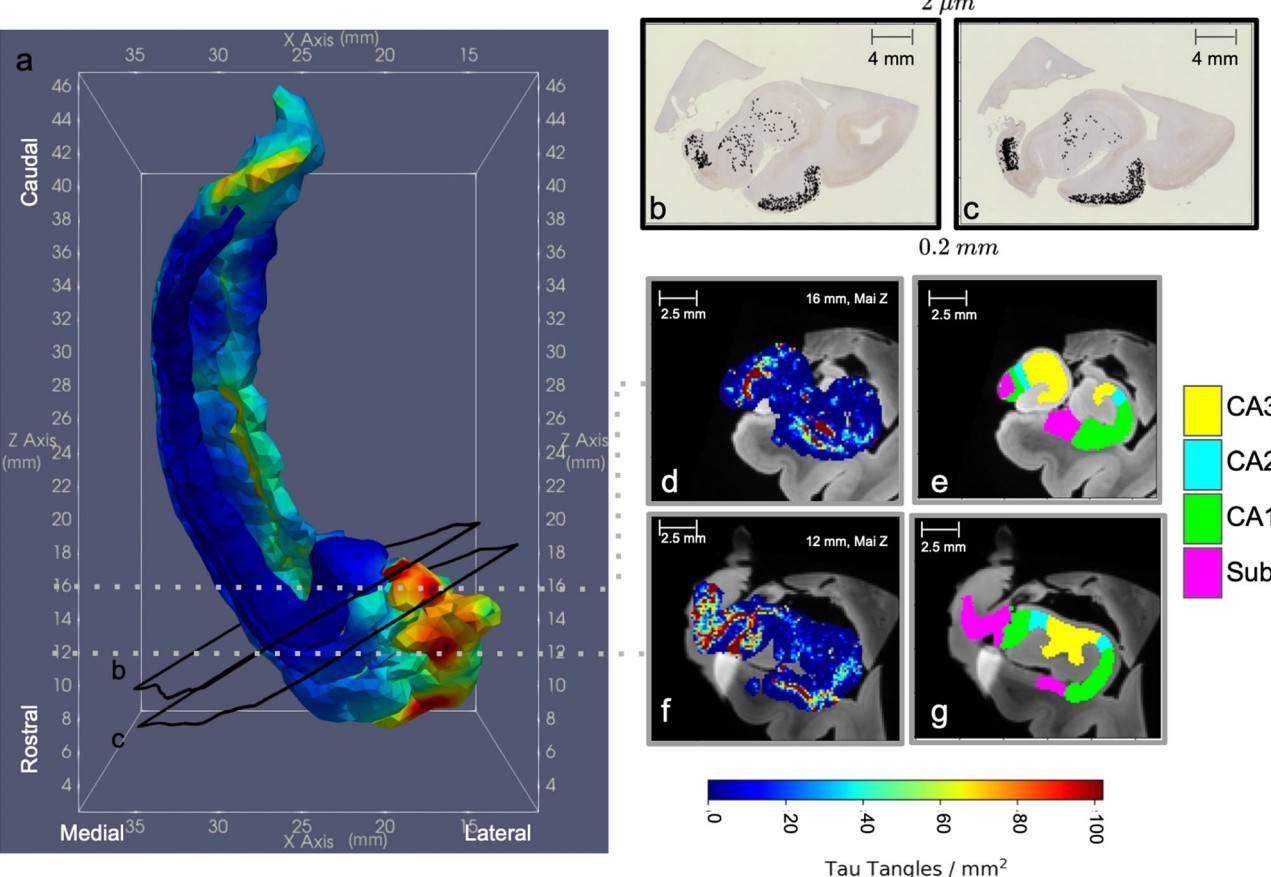

**Fig. 7 Distributions of neurofibrillary tau tangle (NFT) density reconstructed in 3D Mai atlas coordinates. a** Smoothed NFT density measures over the surface of hippocampal subregions (CA1, CA2, CA3, Subiculum (Sub)) generated via the Laplace-Beltrami operator. **b**, **c** Select histology sections in the rostral part of the hippocampus with detected NFTs plotted in black and corresponding position shown in 3D reconstruction. **d–g** Slices of 3D NFT distribution and corresponding MRI with manual segmentations taken at coordinates corresponding to 12 and 18 mm along the z axis defined by the Mai atlas.

The methods presented here constitute a mode of studying not just AD but many other clinical conditions with emerging mechanisms and signatures at both microscopic and macroscopic scales. In Huntington's disease, our group reported early shape changes detectable in MRI[44] that have yet to be linked to precise patterns of inclusion bodies and accumulation of mutant huntingtin protein, as might be observed in histological stains. Others have reported observable changes in various types of MRI (T1, T2, DTI, iron-sensitive) in early (prodromal) stages of Parkinson's Disease and related conditions[45]. Distinguishing related conditions (multiple system atrophy, Lewy body dementia, progressive supranuclear palsy) at early stages of disease is crucial for disease management but requires correlation of MRI signatures to underlying microscopic features seen in histological images. The ability of our methods to map sparse to dense images coupled with the ability to cross image scales and contrasts, particularly in a setting of limited data, enables the addressing of not just these clinical questions but the many remaining gaps in biological understanding at molecular versus tissue scales.

## Methods

**Projective LDDMM algorithm with in-plane transformation.** To solve the Variational Problem 2 for a single 3D atlas, $I_{temp}$, and a set of 2D targets ($J_n$ for $n = 1, \ldots, N$) we formulate an algorithm that alternately optimizes for the deformation in 3D space and the geometric transformation in 2D space, while holding the other fixed. The algorithm can be implemented to incorporate increasing complexity as needed first for crossing modalities and second for crossing resolutions, for instance, to map 3D MRI to 2D histological slices, as is presented here. In its simplest form, $I_{temp}$

and $J$ are of the same modality, yielding targets $J_n$ without expansion, and $\phi_n$ modeled as a rigid motion in plane, as in Lee et al.[4], mapping histological sections to an atlas of the mouse brain. Transformations $\varphi, \phi_n$ for $n = 1, \cdots, N$ are estimated following Algorithm S.1 (Supplementary Note 6).

When $\phi_n$s are broadened to non-rigid diffeomorphisms, as in Tward et al.[5], each $\phi_n$ is estimated in step B via a separate iteration of LDDMM for each target $J_n$. Here, $\phi_n$s encapsulate both rigid and non-rigid components. Separate gradient-based methods are used to update each component in step B with velocity fields updated using Hilbert gradient descent as in ref. [46] and linear transform parameters updated by Gauss-Newton[47].

**Linear prediction algorithm for crossing modalities with scattering transform.** Crossing modalities at similar resolution (e.g., 3D MRI and downsampled 2D histology slices) requires a mapping between range spaces of template and target, giving a similar formulation to that used in Tward et al.[5]. In this work, we introduce the Scattering Transform[16] for crossing modalities at *differing* resolution whereby we build a predictive basis of images from our targets. We model contrast variations between histology and MRI by expanding the color space of our observed set of histology images $\{J_n, n = 1, \cdots, N\}$ from three RGB dimensions to 48 feature dimensions reflecting local radiomic textures of histological scales.

Histology images are each resampled at the resolution of MRI to 48 filtered images, $(S^1_{J_n}, \ldots, S^{48}_{J_n})$ via Mallat's Scattering Transform[16] (see Supplementary Note 3). Unlike a traditional wavelet transform, the Scattering Transform introduces nonlinearities in that the filtration bases are determined by the histology images themselves, which expand the set of textural features captured. For each histological image, a 48-dimensional basis reflective of the tissue characteristics in the actual section itself is used for filtration. We select a common 6-dimensional subspace of these 48 feature dimensions using PCA, yielding a separate basis of six discriminative filtered images, $(\psi^1_n, \ldots, \psi^6_n)$ for each histology slice, $n = 1, \ldots, N$, plus a constant image $\psi^0_n$. We predict the MRI contrast of our transformed projection as a linear combination of these basis elements $\{\psi^j_n\}_{j \in \{0, \ldots, 6\}}$ for each

section $n = 1, \ldots, N$ with linear predictor:

$$J_n^\alpha(\cdot) := \sum_{k=0}^{6} \alpha_n^k \psi_n^k(\cdot) = \alpha_n^T \psi_n(\cdot), \; n = 1, \ldots, N \quad (9a)$$

$$\text{with } \alpha_n := \begin{pmatrix} \alpha_n^0 \\ \cdot \\ \cdot \\ \cdot \\ \alpha_n^6 \end{pmatrix}, \; \psi_n(\cdot) := \begin{pmatrix} \psi_n^0(\cdot) \\ \cdot \\ \cdot \\ \cdot \\ \psi_n^6(\cdot) \end{pmatrix}. \quad (9b)$$

Supplementary Fig. S1 shows a mean-field section using the Scattering Transform. These linear weights $\alpha_n$ are estimated from initialized $\phi_n$s and $\varphi$ following Algorithm 2 (see Supplementary Note 6). Initializations of $\phi_n$ and $\varphi$ are estimated following the approach in Tward et al.[19] in which cubic polynomials are used to match MRI range space to histology range space, generating solutions for $\alpha_n$ via the pseudo-inverse:

$$\alpha_n(\varphi, \phi_n) = \arg\max_{\alpha_n \in R^7} -\left\| J_n^\alpha - \phi_n \cdot P_n I \right\|_2^2 \quad (10a)$$

$$= \int_{R^2} K^{-1} \psi_n(y) \left( \phi_n \cdot P_n I(y) \right) dy$$

$$\text{where } K = \int_{R^2} \left( \psi_n(y) \right) \left( \psi_n(y) \right)^T dy, \; I = I_{\text{temp}} \circ \varphi_1^{-1}. \quad (10b)$$

**Projective LDDMM algorithm with crossing modalities**. The steps described in Algorithm S.2 (Supplementary Note 6) can be naturally incorporated into the framework of Projective LDDMM following Algorithm S.3 (Supplementary Note 6). Here, the linear prediction problem is solved using initial conditions for $\phi_n$ and $\varphi$, and optimization of these geometric transformations then follows from a fixed set of estimated $\alpha_n$s.

**Projective LDDMM algorithm with multiple models**. Introduction of multiple models, as described in the "Scattering Transforms for Digital Pathology" subsection in the Results, replaces the matching cost in (7d) with that of (12a). As a result, the iteration in steps B-C of Algorithm S.3 (Supplementary Note 6) are replaced with an iterative algorithm based on the EM algorithm, implying it is monotonic in the cost. The complete data likelihood for each histology plane $n = 1, 2, \ldots N$ as a function of parameters, $\theta = \phi_n$, with $I = I_{\text{temp}} \circ \varphi_1^{-1}$ and $L(\cdot)$ mapping each location $y \in Y$ to the set of labels $\{1, 2, 3\}$ denoting foreground tissue, artifact, and background, is

$$\prod_{y \in Y} \prod_{k=1}^{3} \frac{1}{(2\pi\sigma_k^2)^{r/2}} \left( \exp\left( -\frac{1}{2\sigma_k^2} \left| \mu_k^n(y) - \phi_n \cdot P_n I(y) \right|^2 \right) \right)^{\mathbf{1}_k(L(y))} \quad (11a)$$

$$\text{with } \mu_k^n(y) = \begin{cases} J_n^\alpha(y) & \text{if } k = 1 \\ \mu_A & \text{if } k = 2 \\ \mu_B & \text{if } k = 3 \end{cases}, \quad (11b)$$

where $r$ denotes the dimension of the range space of $I(\cdot)$ and $\mu_A$ and $\mu_B$ represent the means for artifact and background Gaussian distributions. The E-step takes the conditional expectation of the complete data log-likelihood with respect to the incomplete data, and the previous parameters $\theta^{old}$. The M-step generates our sequence of parameters:

$$\text{E} - \text{step } Q(\theta; \theta^{old}) = -\sum_{k=1}^{3} \frac{1}{2\sigma_k^2} \left\| \pi_{n,k}^{\frac{1}{2}} \left( \phi_n \cdot P_n I - \mu_k^n \right) \right\|_2^2 \quad (12a)$$

$$\text{with } \pi_{n,k}(\cdot) = E\left( \mathbf{1}_k(L(\cdot)) | P_n I(\cdot), \theta^{old} \right).$$
$$\text{M} - \text{step } \theta^{new} = \arg\max_{\theta \in \Theta} Q(\theta; \theta^{old}). \quad (12b)$$

The spatial field of weights $\pi_{n,k}(\cdot)$ is the conditional expectation of the indicator $\mathbf{1}_k(L(\cdot))$. The GEM algorithm[23] solves the maximization step: generating a sequence with increasing log-likelihood:

$$Q(\theta; \theta^{old}) < Q(\theta^{new}; \theta^{old}).$$

**Tau pathology detection**. Patterns of tau pathology are summarized as total counts of NFTs per mm$^2$ of cross-sectioned tissue. NFT counts were computed using a 2-step algorithm: (1) prediction of per pixel probabilities of tau and (2) segmentation of these probability maps into discrete NFTs.

As described previously[19], we used a CNN to model and predict probabilities of being part of a tau tangle for each pixel in a digital histology image. To capture larger contextual features as well as local information for producing per pixel probabilities at high resolutions, we trained a UNET[48] with the architecture shown in Supplementary Fig. S2. Training data was generated on every third slice of histology from the single brain sample analyzed here. Between 8 and 24 sample zones, sized 200-by-200 pixels were selected at random until 8 zones covered tissue (not background). Every pixel in each zone was manually annotated, 1 or 0, as part

of a tau tangle or not. Estimates of accuracy in per pixel tau probabilities were computed using 10-fold cross-validation on the entire training dataset. Supplementary Table S1 shows accuracy metrics for each fold, with mean AUC of 0.9860 and accuracy of 0.9729.

Counts of NFTs in each histology slice were generated by segmenting the probability maps output from the trained UNET. Segmentations were computed using an opensource (OpenCV) implementation of the watershed algorithm to extract connected components with high probability of tau. Here, probability maps, as shown in Supplementary Fig. S3, were first thresholded using Otsu's method. The thresholded image is transformed into what looks like a topographic map by computing the distance of each foreground pixel to the background, with valleys considered those regions at the largest distance from background. Connected components are found by simulating the flow of water from the sink of each of these valleys outward until the edges of the water meet. Each valley, given a minimum distance between adjacent valleys, is considered a separate NFT, with calculated center, area, and roundness as features. Supplementary Fig. S3 shows an example 1 mm square of a histology image together with the predicted probability map and detected NFTs following the watershed algorithm.

The results presented here reflect handling of a set of histological images from a single brain sample. Given the variation in staining intensity we've observed between brain samples collected at different times by different technicians, we plan to train a separate UNET for each sample we evaluate with training data exclusively from that sample. Future work to reduce the burden of manual annotation and increase the generalizability of detection across brain samples includes investigating methods of semi-supervised learning and training data amplification with randomly applied transformations to replace or add to our current process of manual annotation for training.

**Particle representation of histological data**. In the following two sections, we detail our means of representing histological data with a measure-based framework over physical space and feature space as introduced in Miller et al.[29]. We denote these measures, $\mu$, borrowing their notation[29], and describe the action of transformations on these measures to bring them to the 3D space of the Mai Paxinos Atlas and consequently our multiresolution resampling of them with a spatial kernel and feature map.

Here, we model histology data at the microscopic scale specifically following a discrete measure framework[29], where each particle of tissue, indexed by $i \in I$, carries a weighted Dirac measure over histology image space and a Dirac measure over the feature space $w_i \delta_{y_i} \otimes \delta_{f_i}, y_i \in Y \subset R^2$ and $\mathcal{F} = R^{2\ell}$. Weights reflect sampled tissue area captured in each particle measure. At the finest scale ($\mu^0$), this is defined as cross-sectional area in the histology plane $w_i \in \{4 \, \mu m^2, 0\}$, computed with thresholding using Otsu's method. The first $\ell$ dimensions of $f_i \in \mathcal{F}$ denote the number of tau tangles in the area of each of $\ell$ MTL subregions captured by the particle. The second $\ell$ dimensions denote the fraction of the particle's total sampled tissue area ($w_i$) within each of $\ell$ MTL subregions. At the finest scale, each particle captures one pixel of information, so the values of these features are 0 or 1, with

$$f_i \in \left\{ \{0, 1\}^{2\ell} \Big| \sum_{j=1}^{\ell} f_i^j \le 1, \sum_{j=\ell+1}^{2\ell} f_i^j \le 1 \right\}. \quad (13)$$

The left image in Fig. 6 illustrates the particle representation for a single histology image with size of particle reflective of weight and color of particle representing the sum of the first $\ell$ features (i.e., the total number of tau tangles captured by the particle).

We transfer measures $\mu_n^0(dy, df) = \sum_{i \in I} w_i \delta_{y_i}(dy) \delta_{f_i}(df)$ via diffeomorphisms $\phi_n$ and $\varphi$ and rigid transformation to the space of the MRI template and the Mai Paxinos Atlas. Discrete weights $w_i$ adjust according to in-plane expansion/contraction of cross-sectional tissue area, as exhibited in Fig. 6, with adjustment at the fine scale by $\phi_n$ given by the varifold action:

$$\phi_n \cdot \mu_n^0(dy, df) = \sum_{i \in I} w_i |d\phi_n| \delta_{\phi_n(y_i)}(dy) \otimes \delta_{f_i}(df). \quad (14a)$$

To cross scales we use the decomposition of the particle measures

$$\mu(dx, df) = \rho(dx)\mu_x(df), \quad (15)$$

with $\rho$ being the density of the model and $\mu_x$ the field of conditional probabilities on the features. Our transformation across scales nonlinearly rescales space and smooths the empirical feature distributions.

Spatial resampling is determined by the function $\pi(x, x')$, which we define here to be the fraction the particle at $x$ has assigned to it from the particle at $x'$, with $\int_{R^d} \pi(x, x') dx' = 1$. The smoothing on the field of conditional probabilities gives the remapped measure $\mu^1$.

Spatial resamplings at MRI resolutions (0.125 mm) and over surface boundaries of MTL subregions were achieved through isotropic Gaussian resampling and nearest neighbor resampling, respectively, through choice of $\pi$ (see Supplementary Note 4). Feature reduction occurs via maps $\beta \mapsto \gamma(\beta) \in \mathcal{F}'$ for probabilities $\beta$:

$$\mu^1 = \int_{R^d} w^1(y)\delta_y \otimes \mu_y^1 \, dy \text{ with } \begin{cases} \rho^1 = \int_{R^d} w^1(y)\delta_y \, dy \\ \mu_y^1 = \delta_{\gamma(\alpha_y)} \end{cases}. \quad (16)$$

Here, $\gamma(\cdot)$ reduces feature dimension by taking empirical distributions $\alpha_y$ over each of $2\ell$ dimensions to expected first moments for each corresponding

dimension, giving $\mathcal{F}' = R^{2\ell}$ with:

$$\gamma(\beta) := \left( \int_{\mathcal{F}} f_j \beta(df) \right)_{1 \le j \le 2\ell}. \tag{17}$$

Total NFT density is computed from the sum of the first $\ell$ features while NFT density per region is computed from the ratio of feature value $j$ to $\ell + j$ for any of $j = 1, \cdots, \ell$ MTL subregions.

**Surface smoothing with Laplace-Beltrami operator**. Spatial variations in NFT density within MTL subregions are visualized as smooth functions over the surface of each corresponding region. Triangulated mesh surfaces of each subregion are first constructed from manual segmentations using restricted Delauney triangulation, where vertices lie exactly on segmentation boundaries, and where minimum edge length of a triangle is ~6 voxels wide (0.75 mm). Particle mass belonging to a given subregion volume is projected to the surface boundary using a nearest neighbor kernel for $\pi(\cdot, \cdot)$, as defined in the "Particle Representation of Histological Data" subsection in the Methods:

$$\pi(x, x') := \begin{cases} 1 & \text{if } x' = \arg\min_{X'} \|x - x'\|_2^2 \\ 0 & \text{otherwise} \end{cases}. \tag{18}$$

We construct functions, $g_r(x)$ and $g_a(x)$, to represent the total number of NFTs and cross-sectional area of tissue from discrete particle measures of particles projected to the surface vertices $x_i \in V$. To generate smooth representations of NFT density $(\frac{\hat{g}_r(\cdot)}{\hat{g}_a(\cdot)})$, we build a complete orthonormal basis on each curved manifold using the Laplace-Beltrami operator[49]. We treat each MTL subregion independently, generating a separate basis for the closed surface boundary corresponding to each subregion. The resulting basis functions (harmonics) encapsulate the specific architecture of each surface. Supplementary Fig. S4 illustrates the first four basis functions computed for the CA1 surface. As in the Fourier basis, the first harmonic with eigenvalue of 0 yields a constant value representative of overall surface shape. Following computation of the basis elements for each structure, we expand each of the functions $g_r(\cdot)$ and $g_a(\cdot)$ in this basis:

$$\begin{aligned} \hat{g} &= \arg\min_{\hat{g}} \|\hat{g} - g\|_2^2 + k\|\nabla\hat{g}\|_2^2 \\ &= \sum_{i=1}^{N} \frac{\langle g, \beta_i \rangle_V \beta_i(\cdot)}{1 - k\lambda_i w(\cdot)}, \text{ with } \langle g, \beta_i \rangle_V := \sum_{y \in V} \beta_i^*(y)g(y)w(y) \end{aligned} \tag{19}$$

for both $g = g_r(\cdot)$ and $g = g_a(\cdot)$, where $\mathcal{B} := \{\beta_1, \cdots, \beta_N\}$ is a basis for the Laplace-Beltrami operator, and $k$ the smoothing constant (see Supplementary Note 5). We use these bases for smoothing over each corresponding surface in lieu of the standard Euclidean basis for $R^3$, for which the Laplacian operator yields a basis of sines and cosines. As a result, both tissue and NFT mass for each subregion are smoothed according to the architecture of the given subregion.

**Specimen Preparation and Imaging**. The brain tissue used in this study was obtained through postmortem donation to the Johns Hopkins Brain Resource Center. The sample came from a 93-year-old male who had been clinically diagnosed with dementia of the Alzheimer's type. Neuropathologic staging characterized the sample as having high levels of AD pathologic change with Braak stage VI NFT pathology, a CERAD neuritic plaque score of B[27], and no TDP-43 pathology. From the formalin immersion fixed brain, a portion of the MTL including ERC, amygdala, and hippocampus, was excised in 3 contiguous blocks of tissue, sized 20-30 mm in height and width, and 15 mm rostral-caudal (see reconstructed MRI of tissue blocks, Fig. 3).

Each block was imaged with an 11T MR scanner at 0.125 mm isotropic resolution. MR images were b0 diffusion-weighted images and were bias-corrected using a method related to N3. Like N3, we iteratively estimate a smooth field that when multiplied against our image, maximizes specifically the high frequencies in the distribution of tissue intensities within the image[50], but we use Entropy as a measure of high-frequency content, rather than a convolution-based approach. Each block was then cut into two or three sets of 10-micron thick sections, spaced 1 mm apart. Each block yielded between 7 and 15 sections per set. Sets of sections were stained with PHF-1 for tau tangle detection, 6E10 for Aβ plaque detection, or Nissl. Sections stained with PHF-1 or 6E10 were counterstained with hematoxylin. All stained sections were digitized at 2-micron resolution.

**Segmentations of MTL subregions**. MTL subregions within the hippocampus were manually delineated using Seg3D software. Individual block MRIs were rigidly aligned using an in-house manual alignment tool, and per voxel labels were saved for the composite MRI for each brain. Delineations were deduced from patterns of intensity differences, combined with previously published MR segmentations[51,52] and expert knowledge on the anatomy of the MTL. The established borders were applied in three other brains, showing consistent results (in preparation, Eileen Xu, Claire Chen, Susumu Mori, DT, Juan Troncoso, Alesha Seifert, Tilak Ratnanather, Marilyn Albert, MW, and MM). In the brain sample examined here, corresponding regional delineations were drawn on all histology sections stained with PHF-1 (see Fig. 5). Delineations were based on visible anatomical markers and were afterwards confirmed with a corresponding Nissl-stained set of sections. In each of these sections, cytoarchitectonic borders between areas of the hippocampus were indicated, independently from the other datasets, using

previously published cytoarchitectonic accounts of the MTL[53-55]. Labels were assigned per pixel to 4x-downsampled histology images at a resolution of 32 microns and used to evaluate accuracy of registration (see the "Integration of Tau Imaging Data into Multiscale 3D Maps" subsection in the Results). Particular regions of interest include cornu ammonis fields (CA1, CA2, CA3), and subiculum (see Fig. 3).

**Reporting summary**. Further information on research design is available in the Nature Portfolio Reporting Summary linked to this article.

## Data availability

All imaging data (MRI and digital pathology) analyzed in this study is available upon request. Please contact Kaitlin Stouffer (kstouff4@jhmi.edu) for requests.

## Code availability

Code used for training and applying UNET in tau tangle detection can be found here: https://github.com/twardlab/ADproject. Code for solving Projective LDDMM, building measure theoretic data representations, and resampling across scales can be found here: https://github.com/kstouff4/projective-lddmm.

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

## Acknowledgements

This research was funded in part by grants from the National Institutes of Health (U19-AG033655, P30-AG066507, P41-EB031771, R01-EB020062 (M.M.), T32-GM136577 (K.S.), U19-MH114821, R01-NS074980-10S1, RF1MH126732, RF1MH128875, RF1MH28888 (D.T.)), the Kavli Neuroscience Discovery Institute (M.M., D.T.), and the Karen Toffler Charitable Trust (D.T.).

## Author contributions

K.S, M.M., and D.T. designed and developed methods. K.S. and D.T. implemented algorithms. M.W. contrived the method for and oversaw the manual segmentation of histology images and MRI. K.S. and M.M. drafted manuscript. All authors contributed to editing the final manuscript.

## Competing interests

The authors declare no competing non-financial interests but the following competing financial interest: M.M. is an owner of Anatomy Works with the arrangement being managed by Johns Hopkins University in accordance with its conflict of interest policies.

## Ethics

This work was deemed not Human Subjects Research by the Johns Hopkins University IRB.
