## [Peer Review File · Communications Engineering]

Reviewers' comments:

Reviewer #1 (Remarks to the Author):

In this manuscript, the authors describe a method to perform registration of 3D MRI to 3d histological images. This task presents challenges, particularly when registering radiological data, which has relatively lower spatial resolution (mm scale), with tissue sections stained for histopathological markers (micron scale), which show "ground truth" pathology but are susceptible to data loss due to inevitable folding and tearing during tissue processing. The subject matter is relevant to performing radiological-pathological correlations in both animal and human tissues to determine whether MRI is a suitable method to detect pathological markers of neurodegeneration. The authors have applied their method to assess phosphorylated tau tangles in tissue specimen with a diagnosis of Alzheimer's Disease. The literature review in the introduction is thorough and provides suitable background information. The authors have also quantitatively assessed the quality of registration. I do feel that the limitations of the work should be more clearly addressed in the discussion, specifically regarding the following:

1. The phosphorylated tau pathology assessed in this manuscript is from severe Alzheimer's Disease, where the NFTs are widely distributed across cortical layers. In the exemplar images show, this provides excellent gray/white matter contrast, even without the use of a counterstain. How effective would the registration method be in other tauopathies (such as chronic traumatic encephalopathy) where the histopathological markers of interest are more focally distributed. Would a counterstain providing similar contrast properties with the T2 MRI be necessary to produce robust registration results? The authors have used a myelin stain and Nissl stain in adjacent sections, but my question is regarding the ptau pathology.
2. How robust is the registration algorithm to MRI methods such as diffusion MRI, where the MR images are susceptible to distortion, even compared to a T2 weighted scan?

I would recommend that the discussion should be revised (minor revisions) to better reflect the limitations of the work.

Reviewer #2 (Remarks to the Author):

Key results:

This manuscript described a diffeomorphic mapping work (the authors call it "Projective LDDMM") to reconstruct a dense 3D MRI atlas from a set of 2D histology sections based on a single ex-vivo brain medial temporal lobe (MTL) sample. This Projective LDDMM method applies several techniques, as the authors earnestly quoted and referenced. The paper has good contents and they are well assembled.

Originality and significance:

This work is basically a variation of the authors' previous work. The authors should discuss the groundbreaking novelties of this work compared to the authors' previous work, especially reference [27]. The claims in the discussion section are not strong. These methods may be of interest to other researchers. The authors should make improvement in order for this paper to be published.

Data & methodology:

The data quality of the work is solid. However, the presentation of the methodology needs to be improved, see comments below.

General comments:

- As the authors termed that this paper would influence thinking in the area such as biomedical engineering, health sciences, and medical imaging, authors should bear in mind the wide range of researchers in these areas when preparing the materials to describe this work. Although the

abstraction of mathematics is important, the intuitive operational diagrams and figures are also important. For example, many MRI processing works apply wavelet transform, but few present it in pure abstract mathematics. Even in S.3., despite the mathematical equations, the actual applied wavelet form was not presented. Moreover, the diffeomorphic mapping for co-registration of non-rigid images, widely used in the medical image works, were normally presented with additional intuitive forms by researchers, including in authors' work [27].

As an example of comment above, Figure 1 is no different from Equation 1. One suggestion is to expend the Figure 1 into a "Projective LDDMM" process diagram figure with images to represent each step of the process similar to Figure 2. This would easily broaden the readership and the readers would take less time to understand the purpose and working methods of the work.

- The authors should discuss the clinical application of the method.
- The authors should discuss the advantage and significance of the MRI based dense anatomical coordinates. The authors can discuss the possibility of directly imaging this tissue contract with adequate MRI RF pulse sequences.
- The authors discussed the detection of tau tangle in digital histology images using CNN and UNET based on the network trained from partial digital histology images. The authors should perform an experiment and discuss the effectiveness of the network on other individual AD brain samples, and the possibility of replacing manual marking of millions of pixels on every individual histology image.
- The authors manually segmented the MRI and computer reconstructed the MTL subregions. The authors should discuss the accuracy of the reconstructions compared to the manual segmentations.

Other comments:

Line 069: The LDDMM should be stated at least once as "Large Deformation Diffeomorphic Metric Mappings (or Matching)", although it is a well-known methodology term.

Line 323: Section 2.3. 2D PET and Parallel Beam Tomography are not clinically relevant. The authors should consider removing it or replacing it.

Line 737: Table 1. The authors used many pure abstraction mathematics in the paper, but detailed here every Conv and ReLU in the network. This is unnecessary. The authors could use an UNET diagram with image size of each network layer like other common presentations.

Line 775: Instead of only mentioning reference [54], the watershed algorithm should be briefly described, and with an intuitive and meaningful flowchart/ schematic.

Line 783: Sections 4.6, and possibly 4.7, can be accompanied by an image or diagram for better clarity and intuitiveness.

Dear Reviewers,

We thank you for your careful consideration of our manuscript and your poignant suggestions for improvement. We have reproduced your comments below in black text. Our responses to each comment are shown in **green text**, and we indicate where and what corresponding changes we have made in the manuscript with cited line numbers and **blue text**. These changes are copied directly from our source latex document and therefore might reflect particular references to equations, figures, and sources. For bibliographic sources we have since added to the original manuscript, we explicitly cite these in blue text. For those sources originally cited in the manuscript, we do not reproduce those directly here and instead refer to them with their respective codes within the revised snippets of text shown here in blue. We have highlighted the changes to the manuscript with **red text** in the overall manuscript file. Thank you again for your time and consideration.

Reviewer 1 Comments

In this manuscript, the authors describe a method to perform registration of 3D MRI to 3d histological images. This task presents challenges, particularly when registering radiological data, which has relatively lower spatial resolution (mm scale), with tissue sections stained for histopathological markers (micron scale), which show “ground truth” pathology but are susceptible to data loss due to inevitable folding and tearing during tissue processing. The subject matter is relevant to performing radiological-pathological correlations in both animal and human tissues to determine whether MRI is a suitable method to detect pathological markers of neurodegeneration. The authors have applied their method to assess phosphorylated tau tangles in tissue specimen with a diagnosis of Alzheimer’s Disease. The literature review in the introduction is thorough and provides suitable background information. The authors have also quantitatively assessed the quality of registration.

We thank the reviewer for this apt summary of the manuscript.

I do feel that the limitations of the work should be more clearly addressed in the discussion, specifically regarding the following:

1. The phosphorylated tau pathology assessed in this manuscript is from severe Alzheimer’s Disease, where the NFTS are widely distributed across cortical layers. In the exemplar images show, this provides excellent gray/white matter contrast, even without the use of a counterstain. How effective would the registration method be in other tauopathies (such as chronic traumatic encephalopathy) where the histopathological markers of interest are more focally distributed. Would a counterstain providing similar contrast properties with the T2 MRI be necessary to produce robust registration results? The authors have used a myelin stain and Nissl stain in adjacent sections, but my question is regarding the ptau pathology.

We thank the reviewer for their questions and the opportunity to comment further on the robustness of our method in the context of other neuropathologies and staining protocols. We first want to clarify that in the digital histological images examined in this manuscript, those

stained for ptau with PHF-1 were counterstained with hematoxylin, which inherently does distinguish features of gray/white matter in highlighting neuronal cell bodies. While we have not tested our approach with histological sections stained with an alternative counterstain, we believe our approach should be robust enough to accommodate a range of counterstains. The Scattering Transform we use to build a basis of images to predict MR contrast extracts many texture-based dimensions at multiple scales from histological images, which we imagine would be sufficient to predict MR contrast in the context of other counterstains. Nevertheless, we look forward to confirming this hypothesis with the testing of our method on histological images with different counterstains in the future.

Prompted by the reviewer's comment, we have added the following text to our methods section 4.8:

Line 1078-1079: Sections stained with PHF-1 or 6E10 were counterstained with hematoxylin.

Second, while we do examine a case of severe AD, where, as the reviewer notes, NFTs are likely to be distributed across cortical layers, we expect the registration method would perform well in other tauopathies where tau would not be distributed across cortical layers. Even in our single brain sample analyzed, NFTs were not densely found across the spectrum of histological sections from anterior to posterior, and we observed that the registration method worked equally well in these histological slices with little to no NFTs. Nevertheless, future work is needed to confirm the robustness of the method in additional brain samples with different types of pathology, and we look forward to seeing our method applied to the analysis of samples from such conditions as chronic traumatic encephalopathy or primary age-related tauopathy. We address this point and the previous in our discussion with the following added to the manuscript:

Line 558-578: The power of the Scattering Transform is in its generation of filtrations across many scales, which are appropriate given the micro-scale histology which supports them and which are rich enough to build linear predictors to contrasts such as that in MRI. We demonstrate, here, the results of mapping to MRI histological slices with a single staining protocol (PHF-1 counterstained with hematoxylin) and from a case of advanced AD. In part, the distribution of NFTs across cortical layers and the highlighted cell bodies resulting from the hematoxylin stain may provide an advantage to other counterstains and pathologies in successfully mapping to MR contrast that similarly distinguishes gray from white matter. However, we anticipate our methods' success in the context of other pathologies and stains given the robustness of the Scattering Transform for capturing a range of textural features that should allow prediction of any imaging contrast. Furthermore, the successful registration of histological sections from both rostral and central regions of the hippocampus, here, which harbored very different levels of NFT pathology suggests our method could be used to reconstruct digital pathology images from various staining protocols and with diverse patterns and presences of pathology in a 3D MR atlas. We are currently investigating our method in 6E10 stained sections highlighting A β pathology and hope to extend this work to examining additional tauopathies in the future.

2. How robust is the registration algorithm to MRI methods such as diffusion MRI, where the MR images are susceptible to distortion, even compared to a T2 weighted scan?

We thank the reviewer for their question and the opportunity to clarify our methods further. First, we note that the MR images examined here are diffusion MRI (b0 images with a contrast similar to T2) and therefore, might be susceptible to the distortion the reviewer mentions for diffusion MRI. No post-processing procedures were applied specifically to remove distortion. Entropy based bias correction was applied to the MR image for computing alignment, here, with histology images. The code for this bias correction can be found in our repository: <https://github.com/kstouff4/projective-lddmm>. The precise script used is entitled biasCorrect.py.

We have added this clarification to the manuscript in Section 4.8 with the following:

Lines 1053-1057: *MR images were b0 diffusion weighted images and were bias-corrected using a method related to N3 \cite{DEBOOR197250}. Like N3 we use “an iterative method and seek a smooth multiplicative field that maximizes the high frequency content of the distribution of tissue intensity” (as described in \cite{Vovk2007}), but we use Entropy as a measure of high frequency content, rather than a convolution based approach.*

Furthermore, while we have not tested our specific method for aligning histology slices to different types of MR images, the community (including ourselves) has looked extensively at the application of LDDMM to DTI types of images (Cao 2005, Cao 2006, Ceritoglu 2007). We anticipate Projective LDDMM as described here can be applied similarly to MR images of different types and contrasts.

In response to this comment, we have added three references to our manuscript describing past work in LDDMM with DTI imaging. These are found in Section 2.1:

Line 138: *For different problems of interest, the atlas image is R^r -valued with, for instance, $r = 1$ corresponding to single contrast MRI or $r = 6$ for diffusion tensor images (DTI) \cite{Tournier2011}, as mapped with LDDMM in prior work \cite{Cao2005,cao2006diffeomorphic,ceritoglu2009multi}.*

Cao, Y., Miller, M.I., Winslow, R.L., Younes, L.: Large deformation diffeomorphic metric mapping of vector fields. IEEE Transactions on Medical Imaging 24(9), 1216–1230 (2005).

Cao, Y., Miller, M.I., Mori, S., Winslow, R.L., Younes, L.: Diffeomorphic matching of diffusion tensor images. In: 2006 Conference on Computer Vision and Pattern Recognition Workshop (CVPRW'06), pp. 67–67 (2006). IEEE.

Ceritoglu, C., Oishi, K., Li, X., Chou, M.-C., Younes, L., Albert, M., Lyketsos, C., van Zijl, P.C., Miller, M.I., Mori, S.: Multi-contrast large deformation diffeomorphic metric mapping for diffusion tensor imaging. Neuroimage 47(2), 618–627 (2009).

Additionally, we have added the following suggestion in our discussion with a reference to other high field MR image types:

Line 602-607: *In the future, we aim to use our methods in the context of these imaging technologies and investigate the reconstruction of diverse biological measures in a range of MR atlases, from in vivo T2 and DTI to more specialized and higher resolution MRI derived using different RF pulse sequences \cite{POSER2018101} that might afford even higher accuracy of alignment to histological images.*

Benedikt A. Poser, Kawin Setsompop. Pulse sequences and parallel imaging for high spatiotemporal resolution MRI at ultra-high field. NeuroImage, Volume 168, 2018.

I would recommend that the discussion should be revised (minor revisions) to better reflect the limitations of the work.

We believe we have addressed the limitations raised by the reviewer in the discussion as suggested (see above for specifics) and in tandem with the suggestions from reviewer 2 (see below).

Reviewer 2 Comments

Key results:

This manuscript described a diffeomorphic mapping work (the authors call it “Projective LDDMM”) to reconstruct a dense 3D MRI atlas from a set of 2D histology sections based on a single ex-vivo brain medial temporal lobe (MTL) sample. This Projective LDDMM method applies several techniques, as the authors earnestly quoted and referenced. The paper has good contents and they are well assembled.

Originality and significance:

This work is basically a variation of the authors’ previous work. The authors should discuss the groundbreaking novelties of this work compared to the authors’ previous work, especially reference [27]. The claims in the discussion section are not strong. These methods may be of interest to other researchers. The authors should make improvement in order for this paper to be published.

We thank the reviewer for their inquiry into how this work compares to our previous work and the suggestion to strengthen our claims in the discussion section. We believe this is the first time an image model has been formulated that accommodates *both* surjective transformations and their generalization to optical technologies, as captured in our projection operator, *and* the crossing of scales and contrasts, as enabled by our construction of a nonlinear basis of images from the Scattering Transform. We anticipate its formulation, here, as a unified optimization problem that can be applied to more diverse imaging sets not only from digital pathology and MRI, but a range of imaging types across scales. While the polynomial transformation from

Tward [27] accommodates a transformation of one modality to another at *low resolution*, scattering transforms present a general method for expanding the orbit model of CA to accommodate different *high resolution* radiomic textures of imaging modalities.

In addition to the presentation of this unified formulation, we have introduced, here, added dimensions of in-plane nonlinear deformation to be estimated for each target image that accommodate a broader range of distortion seen in histological processing and emerging methods in spatial transcriptomics. Additionally, we have used a measure-based framework of data modeling that enables the construction and resampling of data distributions across scales and both within volumes and over surface boundaries of structures of interest. As presented here, this method affords better opportunity for integration and comparison of molecular measures with macroscopic shape markers, as is particularly relevant in the study of AD and other diseases.

As per the reviewer's suggestion to strengthen our discussion, we have adjusted our discussion with the following:

Lines 579-607:

We believe this is the first time an image model has been presented that simultaneously accommodates the generation of sparse or partial image captures from a dense image atlas and an expanded image representation for crossing scales and contrasts. The key technology used for crossing scales in this work, without losing high resolution information, is the Scattering Transform. We have demonstrated the success of this image model at capturing the surjective measures of digital histology images stained with PHF-1 for NFT detection from 3D MRI. However, as highlighted in \ref{petSec} and \ref{ctExample}, this formulation applies to a range of established imaging technologies from classical parallel beam projection tomography to optical sectioning and emerging imaging technologies, such as light sheet methods and those in spatial transcriptomics.

In addition, we have introduced, here, two tactics to further increase the robustness of our methods to this range of imaging modalities and preparations. We have demonstrated a new class of Gaussian mixture models for modeling the random effects of tears and histological processing, each occurring independently in the histology coordinates per section. The mixtures are built to expand the orbit of images generated by the Scattering Transform by including non-foreground tissue as manifest by the smooth deformation of the MR template. Additionally, the introduction of a high dimensional in-plane geometric transformation for each target histological slice departs from previous work \cite{Lee2018, Lee2021, tward2020}. These added dimensions accommodate the higher levels of distortion that might arise in complex staining and processing procedures such as those in histopathology and spatial transcriptomics. In the future, we aim to use our methods in the context of these imaging technologies and investigate the reconstruction of diverse biological measures in a range of MR atlases, from in vivo T2 and DTI to more specialized and higher resolution MRI derived using different RF pulse sequences \cite{POSER2018101} that might afford even higher accuracy of alignment to histological images.

We have also emphasized these contributions in the introduction with the following alterations:

Lines 68-72: *The major contribution of this work is the introduction of a new class of image-based diffeomorphism methods which we term Projective Large Deformation Diffeomorphic Metric Mapping (Projective LDDMM) for aligning sparse sets of image captures to 3D coordinate systems across micron and millimeter scales.*

Lines 94-97: *We previously used a polynomial transformation to accommodate crossing modalities at a single (low) resolution \cite{tward2020}. The second major contribution of this work is the introduction of a novel photometric transformation of histology to MRI without loss of high resolution information.*

Data & methodology:

The data quality of the work is solid. However, the presentation of the methodology needs to be improved, see comments below.

General comments:

- As the authors termed that this paper would influence thinking in the area such as biomedical engineering, health sciences, and medical imaging, authors should bear in mind the wide range of researchers in these areas when preparing the materials to describe this work. Although the abstraction of mathematics is important, the intuitive operational diagrams and figures are also important. For example, many MRI processing works apply wavelet transform, but few present it in pure abstract mathematics. Even in S.3., despite the mathematical equations, the actual applied wavelet form was not presented.

We thank the reviewer for their suggestion for operational diagrams and figures. We address this suggestion further below.

We thank the reviewer for drawing a comparison between our wavelet-based processing and traditional MRI wavelet processing techniques, and we welcome the opportunity to clarify how our methods compare and differ to those in other MRI processing pipelines. While the Scattering Transform shares many features with linear wavelet processing, particularly its multi-scale nature, it is instead nonlinear as it incorporates a number of modulus operations together with wavelet transforms. Additionally, as described in Section 4.2, we use the Scattering Transform to build a basis of filtered images from which we predict a corresponding MR contrast image. It strongly departs from linear basis methods in that the basis of image filters are not fixed but are generated from the histological images themselves.

In response to this comment, we have included the following sentences in section 4.2 in the manuscript to emphasize this comparison:

Line 712-716: *Unlike a traditional wavelet transform, the Scattering Transform introduces nonlinearities in that the filtration bases are determined by the histology images themselves, which expand the set of textural features captured. For each histological image, a 48-*

dimensional basis reflective of the tissue characteristics in the actual section itself is used for filtration.

Additionally, as we describe in S.3, we use a Gaussian high pass filter in place of what is specifically a wavelet in order to achieve rotation invariance of our transform. We thank the reviewer for highlighting our omission of this exact filter in the manuscript and have included it as suggested.

Line 1252-1255: We use 16 paths, $p_i \in P$ of length 1 or 2 and a high pass Gaussian filter, $(1 - \frac{1}{2\pi\lambda_i} \exp(-\frac{x^2 + y^2}{2\lambda_i^2}))$, with width dilated according to λ_i , in place of a traditional wavelet to achieve a representation both translation and rotation invariant in addition to Lipschitz continuous to small deformations.

Moreover, the diffeomorphic mapping for co-registration of non-rigid images, widely used in the medical image works, were normally presented with additional intuitive forms by researchers, including in authors' work [27].

As an example of comment above, Figure 1 is no different from Equation 1. One suggestion is to expand the Figure 1 into a "Projective LDDMM" process diagram figure with images to represents each step of the process similar to Figure 2. This would easily broaden the readership and the readers would take less time to understand the purpose and working methods of the work.

We thank the reviewer for their suggestion. We have altered our Figure 1 accordingly, as seen in the manuscript in Section 2.1. Additionally, we have added the following to the Figure 1 caption to reflect these changes:

Lines 172-175: Example images shown for each step in the model: 3D MRI template (left), projection slices of deformed MRI template (middle), and estimated MR contrast of digital histology image (right). Post sectioning parameters capture observed changes in sulcal width and ventricle shape in histological sections (right) compared with idealized MRI projections (middle).

- The authors should discuss the clinical application of the method.

We thank the reviewer for their comment and welcome the suggestion to elaborate on the clinical applications of our method. Though we illustrate the application of our method to the linking of microscopic pathology and clinical biomarkers for AD, we believe a similar strategy can be taken in the context of other neurodegenerative diseases where imaging biomarkers have been posed for earlier detection and management but have not been linked to molecular signatures. These include both Huntington's Disease and Parkinson's Disease. Additionally, as formulated, our methods accommodate the mapping of not just histological images but other types of "sparse" or "partial" images to dense atlases. Applications involving the link of emerging

data types such as that of spatial transcriptomics to dense MR atlases are amongst our own investigations in the future.

Accordingly, we have added the following paragraph (with two additional references related to Huntington's Disease and Parkinson's Disease) to our Discussion section.

Lines 638-655: The methods we've presented here constitute a mode of studying not just AD but many other clinical conditions with emerging mechanisms and signatures at both microscopic and macroscopic scales. In Huntington's disease, our group has reported early shape changes detectable in MRI \cite{Younes2012,FARIA2016450} that have yet to be linked to precise patterns of inclusion bodies and accumulation of mutant huntingtin protein, as might be observed in histological stains. Others have reported observable changes in various types of MRI (T1, T2, DTI, iron-sensitive) in early (prodromal) stages of Parkinson's Disease and related conditions \cite{heim2017magnetic}. The distinguishing of these related conditions (multiple system atrophy, Lewy body dementia, progressive supranuclear palsy) at early stages of disease is crucial for management of disease but will require correlation of these MRI signatures to the underlying distinguishing features of these conditions seen in histological images. The ability of our methods to map sparse to dense images coupled with the ability to cross image scales and contrasts enables the addressing of not just these clinical questions but the many remaining gaps in biological understanding at molecular versus tissue scales.

Younes, L., Ratnanather, J., Brown, T., Aylward, E., Nopoulos, P., Johnson, H., Magnotta, V., Paulsen, J., Margolis, R., Albin, R., Miller, M., Ross, C., PREDICT-HD Investigators and Coordinators of the Huntington Study Group: Regionally selective atrophy of subcortical structures in prodromal hd as revealed by statistical shape analysis. Human Brain Mapping 35(3), 792–809 (2014).

Faria, A.V., Ratnanather, J.T., Tward, D.J., Lee, D.S., van den Noort, F., Wu, D., Brown, T., Johnson, H., Paulsen, J.S., Ross, C.A., Younes, L., Miller, M.I.: Linking white matter and deep gray matter alterations in premanifest huntington disease. NeuroImage: Clinical 11, 450–460 (2016).

Heim, B., Krismer, F., De Marzi, R., Seppi, K.: Magnetic resonance imaging for the diagnosis of parkinson's disease. Journal of neural transmission 124(8), 915–964 (2017).

The authors should discuss the advantage and significance of the MRI based dense anatomical coordinates. The authors can discuss the possibility of directly imaging this tissue contract with adequate MRI RF pulse sequences.

We thank the reviewer for their suggestion to discuss the benefits of MRI based dense anatomical coordinates. We believe one of the key benefits of Projective LDDMM is its ability to map sparse to dense images. Here, the mapping of sparsely sampled digital histology images to a high field MRI atlas affords two opportunities. First, it enables us to pool imaging data across a set of histological sections to compute and visualize the 3D distribution of 2D pathological measures (e.g. NFTs per square mm). Our mapping to a high field MRI further enables us to construct these distributions at a much higher resolution than typical in vivo imaging (MRI or

PET) affords. Second, the mapping to a dense MRI metric facilitates the correlation of pathological measures, computed from histology images, to many of the biomarkers such as shape and thickness changes that depend on a dense 3D metric. In the future, we aim to investigate these correspondences between clinical MR atlases and distributions of pathology from additional brain samples.

We have summarized the above discussion in our manuscript by altering and adding the following sentences in our Discussion:

Lines 610-620: Reconstruction within the dense 3D metric of the brain, as defined by our high field MRI atlas, enables both pooling of the pathological measures across sets of histology images and resampling of these measures at micron and millimeter resolution and within 3D volumes and 2D manifolds of interest. The submillimeter resolution of the tau tangle maps we've shown here exceeds that achieved by current technologies in Tau PET. Additionally, the reconstruction of these maps within the dense 3D MRI atlas facilitates correlation with many of the reported biomarker measures such as shape and thickness changes \cite{kulason2019, Younes2019}, defined within such MR atlases. Directly correlating these biomarker measures to reconstructed distributions of pathology at different stages of AD will be the subject of future work.

With regard to the second point, we appreciate the reviewer's inquiry into other types of MRI and believe that more specialized imaging of tissue would offer an even further improvement in accuracy of alignment between histology and MRI as presented here. We note this possibility in our discussion with the addition of the following lines and a reference to the different types of MRI that could be investigated in the future using such different pulse sequences:

Lines 602-607: In the future, we aim to use our methods in the context of these imaging technologies and investigate the reconstruction of diverse biological measures in a range of MR atlases, from in vivo T2 and DTI to more specialized and higher resolution MRI derived using different RF pulse sequences \cite{POSER2018101} that might afford even higher accuracy of alignment to histological images.

Benedikt A. Poser, Kawin Setsompop. Pulse sequences and parallel imaging for high spatiotemporal resolution MRI at ultra-high field. *NeuroImage*, Volume 168, 2018.

- The authors discussed the detection of tau tangle in digital histology images using CNN and UNET based on the network trained from partial digital histology images. The authors should perform an experiment and discuss the effectiveness of the network on other individual AD brain samples, and the possibility of replacing manual marking of millions of pixels on every individual histology image.

We thank the reviewer for their inquiries into the robustness of our neurofibrillary tangle detection platform on other brain samples and the opportunity to clarify and comment on our training procedure. With regard to the first point, as the reviewer notes, we present the results of

both our UNET-based NFT detection algorithm as well as our mapping methods on ~35 histological images from a single brain sample. Ultimately, our goal is to use this technology in the future to examine histological images from a range of brain samples reflective of different stages of AD pathology to reconstruct 3D NFT distributions at each of these stages. Therefore, the effectiveness of our methods in the context of these other samples will be crucial to evaluate and is the subject of future work.

With regard to NFT detection specifically, we've observed pronounced differences in staining intensities between brain samples and within the same brain sample where adjacent slices have been stained by different technicians at different times. Therefore, we intend to train a separate UNET for each brain sample to ensure most accurate detection of NFTs across the set of histology images for each brain analyzed.

Regarding our training procedure, we want to clarify that we select only a small number of regions on a third of all histology images in which to annotate NFTs. We amplify this set of training data by choosing replicates of subregions within these larger regions and adding random rotations to the subregions to account for differences in orientation of NFTs in different areas of the histology images. We are currently exploring ways to amplify this training data even further using synthesis methods to augment the observed data set as well as the possibility of semi-supervised approaches in the future.

We have adjusted Section 2.4 as follows to clarify the specificity of our detections to a single brain sample and our manual annotation of pixels for validation.

Lines 393-398: NFTs were detected using a machine learning-based algorithm (see Section \ref{sec:tauML}) \textcolor{red}{trained and tested to achieve maximum accuracy of detection for the single set of histologically stained tissue slices examined here}. Per pixel accuracy of identifying tau was evaluated with 10-fold cross validation, yielding an average AUC of 0.9860 and accuracy of 0.9729 (see Section \ref{sec:tauML} for individual fold metrics), and final counts of NFTs were validated against a reserved set of regions \textcolor{red}{across 10 separate histology images} manually annotated for NFTs and totaling 25 million pixels.

We have also added the following to Section 4.5 to discuss future additions and alterations to our NFT detection platform and its application to other brain samples.

Lines 904-912: The results presented here reflect handling of a set of histological images from a single brain sample. Given the variation in staining intensity we've observed between brain samples collected at different times by different technicians, we plan to train a separate UNET for each sample we evaluate with training data exclusively from that sample. Future work to reduce the burden of manual annotation and increase the generalizability of detection across brain samples includes investigating methods of semi-supervised learning and training data amplification with randomly applied transformations to replace or add to our current process of manual annotation for training.

- The authors manually segmented the MRI and computer reconstructed the MTL subregions. The authors should discuss the accuracy of the reconstructions compared to the manual segmentations.

We thank the reviewer for their comment and welcome the opportunity to clarify our methods of reconstruction further. We first note that surfaces of MTL subregions as visualized in Figures 4 and 6 are built from manual segmentations of MRI using restricted Delauney triangulation. These surfaces are therefore directly reflective of the manual segmentations from which they are generated. The vertices of the surface lie on the boundary of the manual segmentation, and are defined as a subset of vertices in a marching cubes triangulation. The faces of the surface are larger to introduce smoothing, and are defined such that the minimum edge length of any triangle is 0.75 mm (e.g. within 6 voxels of the segmentation image).

Second, we wish to clarify in this work that we have discussed three separate types of reconstruction. First, we have placed 2D histology coordinates (at 2 micron resolution) in the space of high field 3D MRI coordinates (at 0.125 mm resolution). We measured the accuracy of this alignment by comparing overlap in manual segmentations of hippocampal subregions demarcated directly on 2D histology images versus deformed from 3D MRI onto 2D histology images, as reported in Section 2.4 and Supplementary Note S.2. Second, we have placed transported measures of NFT density from histology images onto surface renderings of MTL subregions manually segmented in 3D MRI by projecting them to the nearest vertex on the boundary. The accuracy of moving measures from volume to surface is measured by the accuracy of the surface generation algorithm, as described above. Finally, we rigidly transport these surfaces to the space of the Mai Paxinos atlas. Surface renderings of the hippocampus were constructed from coronal images in the Mai Paxinos, occurring at every 1.3 mm rostral to caudal. Accuracy of rigid alignment was qualitatively confirmed by comparison of landmarks on Mai surface and AD brain sample (e.g. anterior and posterior poles of hippocampus, folds in hippocampal head), which appear within 3-4 mm of each other. Given sparsity of coronal samples in the Mai atlas, we therefore expect accuracy of measures as reported in the Mai coordinates to be on the order of 2-4 mm resolution.

To clarify this accuracy in the manuscript, we have added/adjusted the following:

Section 4.7, Lines 1003-1006: *Triangulated mesh surfaces of each subregion are first constructed from manual segmentations using restricted Delauney triangulation \cite{cheng2013delaunay}, where vertices lie exactly on segmentation boundaries, and where minimum edge length of a triangle is approximately 6 voxels wide (0.75 mm).*

Section S.4, Lines 1276-1286: *Surface renderings of the hippocampus, amygdala, and ERC in the Mai atlas were constructed from coronal images at every 1.3 mm using restricted Deulaney triangulation \cite{cheng2013delaunay}. We used a manual alignment tool, created in-house, to select optimal alignments between these surface renderings and those of our brain sample, constructed from manual segmentations on 3D MRI. Alignments were qualitatively confirmed by comparison of landmarks on corresponding Mai surfaces and transformed surfaces of our brain*

sample (e.g. anterior and posterior poles of hippocampus and folds in hippocampal head). These appeared within 3-4 mm of each other. Given that coronal sections in the Mai atlas in this area are approximately 1.3mm apart, we expect accuracy of reported Mai coordinates on the order of 2-4mm.

Other comments:

Line 069: The LDDMM should be stated at least once as "Large Deformation Diffeomorphic Metric Mappings (or Matching)", although it is a well-known methodology term.

We thank the reviewer for bringing this to our attention and agree that the LDDMM abbreviation should be mentioned in full at least once. As the reviewer suggests, we replace LDDMM in its first occurrence in line 70 with "Large Deformation Diffeomorphic Metric Mapping":

Line 69-72: This paper introduces a new class of image-based diffeomorphometry methods which we term Projective Large Deformation Diffeomorphic Metric Mapping (Projective LDDMM) for aligning sparse sets of image captures to 3D coordinate systems across micron and millimeter scales.

Line 323: Section 2.3. 2D PET and Parallel Beam Tomography are not clinically relevant. The authors should consider removing it or replacing it.

We thank the reviewer for their attention to our proposed readership and the suggestion to keep the main body of the paper focused on what is clinically relevant. We have moved these two examples to the Supplementary Note S.1, retitling this section "PET and Classical Tomography" and now Section 2.3 just focuses on optical sectioning. As such, we have expanded our discussion on optical sectioning to describe the form of the point spread more completely and compare the treatment of optical sectioning within our framework to that which has been done previously.

We have specifically added two additional references to Section 2.3 and adjusted the text as follows:

Lines 360-383:

Each projection image, $P_n(\cdot)$ on R^3 of \leqref{projection-variation-problem}, is constructed both from components within the n -th plane of focus and the remaining 3D volume. The relative weight of each component is given by the point spread $p_n(y, dx)$, which is a function of the detection process and of the microscope's optics \cite{Preza:92}, such as aperture size, wavelength of light, amplitude of wave on aperture, and aperture orientation \cite{Gibson:89}. In practice, these point spreads can be determined experimentally by imaging the spread of a point-source bead \cite{Joshi:93} in a variety of locations or can be modeled, as in Gibson and Lanni, based on Kirchoff's scalar diffraction laws \cite{Gibson:89}. Similar optical sectioning effects can be seen with emerging technologies in light sheet microscopy

\cite{becker2019deconvolution}, where an anisotropic 3D point-spread governs the construction of each projection image.

Treatment of these imaging technologies within the framework of Projective LDDMM unifies the number of approaches previously taken at reconstructing the underlying image but here, within the setting of introducing an anatomically complex prior distribution as constrained to be within the diffeomorphic orbit of a template. In previous reconstruction models, the projective intensity of \eqref{projection-variation-problem} is taken either as the mean-field of a Poisson process \cite{Joshi:93} or as the mean and variance fields of a Gaussian approximation to the Poisson field \cite{Preza:92}.

In addition to both light sheet microscopy and confocal optical sectioning, classical imaging modalities such as positron emission tomography (PET) and classical tomography (CT) lend themselves to representation within the framework of Projective LDDMM. We illustrate these examples in detail in Supplementary Note \ref{ctExample}.

Preza, C., Miller, M.I., Thomas, L.J., McNally, J.G.: Regularized linear method for reconstruction of three-dimensional microscopic objects from optical sections. *J. Opt. Soc. Am. A* 9(2), 219–228 (1992).

Becker, K., Saghafi, S., Pende, M., Sabdyusheva-Litschauer, I., Hahn, C.M., Foroughipour, M., Jahrling, N., Dodt, H.-U.: Deconvolution of light sheet microscopy recordings. *Scientific reports* 9(1), 1–14 (2019).

Line 737: Table 1. The authors used many pure abstraction mathematics in the paper, but detailed here every Conv and ReLU in the network. This is unnecessary. The authors could use an UNET diagram with image size of each network layer like other common presentations.

We thank the reviewer for their suggestion and the opportunity to present our UNET in a manner more similar to other presentations and more intuitive to our readership.

We have adjusted our manuscript accordingly by replacing Table 1 in Section 4.5 with a UNET diagram that depicts the image sizes of each network layer together with the overall architecture of the network.

Additionally, we have added the following caption to describe the figure:

Structure of UNET trained to detect NFTs. Input depicted at upper left corner with output at far right. Arrow style reflective of operation (3x3 convolutions with stride 1, rectified linear units, 2x2 max poolings, 2x2 transposed convolutions with stride 2, 1x1 convolution with stride 1, softmax, and concatenation with center cropping). Image size at each stage proportional to size of block: height proportional to number of pixels in image and width proportional to number of channels. Input image size of training data points, here, is 132x132 pixels with 3 channels (RGB).

Line 775: Instead of only mentioning reference [54], the watershed algorithm should be briefly described, and with an intuitive and meaningful flowchart/ schematic.

We thank the reviewer for their suggestion to explain in more detail the watershed algorithm to ensure full comprehension of our readers. We have added a figure to section 4.5 to illustrate the connection between the UNET output and the watershed algorithm. We have also added the following sentences to Section 4.5.

Lines 893-903: Here, probability maps, as shown in Figure \ref{watershedFig}, were first thresholded using Otsu's method \cite{otsu}. The thresholded image is transformed into what looks like a topographic map by computing the distance of each foreground pixel to the background, with "valleys" considered those regions at a largest distance from background. Connected components are found by simulating the "flow of water" from the sink of each of these valleys outward until the edges of the water meet. Each valley, given a minimum distance between adjacent valleys, is considered a separate NFT, with calculated center, area, and roundness as features. Figure \ref{watershedFig} shows an example 1 mm square of a histology image together with the predicted probability map and detected NFTs following the watershed algorithm.

Output after each step in NFT detection algorithm. Input image of histology tissue stained with PHF-1 for tau detection (left). UNET predicted per-pixel probabilities of belonging to an NFT (second). Segmentation of connected high probability components following application of the watershed algorithm (third). Each connected component assigned unique label (2 to 80) and considered a separate NFT. Centers of each NFT shown inscribed in red circle with size proportional to size of detected NFT (right).

Line 783: Sections 4.6, and possibly 4.7, can be accompanied by an image or diagram for better clarity and intuitiveness.

We thank the reviewer for their suggestion to accompany these two sections with images/diagrams. To enhance the clarity of each section, we have added both figures and additional text.

In Section 4.6, we have added a figure to depict our representation of histological information as "particles" and the subsequent transfer of this information to the space of 3D MRI. We had hoped to stress three of the main benefits in using a measure-based framework as: 1) the ability to capture both tissue area and measures of pathology to retain a notion both of how much data is available and what the values of the data are, 2) the ability to transport these measures in physical space under the defined action of a diffeomorphism, and 3) the ability to resample physical space and feature space independently to compute summary statistics of interest on pathology data both within volumes and over manifolds. The figure added to Section 4.6 highlights the first of these two whereas we believe Figure 6 in Section 2.4 highlights the last of these benefits more aptly.

The added figure caption reads:

Particle representation for a single histology image before (left) and after (right) transformation to space of 3D MRI. Size of circles proportional to weight of particles equal to area of tissue sampled. Weights change according to $\text{leqref}\{\text{varaction2}\}$. Color of circles reflects total number of NFTs captured by particle, which is unchanged after transformation.

Additionally, we have added the following descriptions in the text of Section 4.6 to describe the added figure.

Lines 940-943: *The left image in Figure $\text{ref}\{\text{partRep}\}$ illustrates the particle representation for a single histology image with size of particle reflective of weight and color of particle representing the sum of the first ℓ features (i.e. the total number of tau tangles captured by the particle).*

Line 946: *Discrete weights w_i adjust according to in plane expansion/contraction of cross-sectional tissue area, $\text{as exhibited in Figure } \text{ref}\{\text{partRep}\}$, with adjustment at the fine scale by ϕ_n given by the varifold action*

For Section 4.7, we have added a figure depicting the basis functions we use for smoothing function values over the surface boundary of each MTL subregion. The corresponding Figure caption reads:

First four basis elements (harmonics) constructed with the Laplace-Beltrami operator shown for CA1 surface. Corresponding eigenvalues are 0, -0.0025, -0.0081, -0.0180.

In addition, we have added the following text to explain our motivation for using this approach of expansion via a Laplace-Beltrami basis.

Lines 1023-1029: *We treat each MTL subregion independently, generating a separate basis for the closed surface boundary corresponding to each subregion. The resulting basis functions (harmonics) encapsulate the specific architecture of each surface. Figure $\text{ref}\{\text{LBharm}\}$ illustrates the first four basis functions computed for the CA1 surface. As in the Fourier basis, the first harmonic with eigenvalue of 0 yields a constant value representative of overall surface shape. Following computation of the basis elements for each structure, we expand each of the functions g_{τ} and g_a in this basis*

Lines 1041-1042: *As a result, both tissue and NFT mass for each subregion are smoothed according to the architecture of the given subregion.*

Reviewers' comments:

Reviewer #1 (Remarks to the Author):

The authors provided thoughtful and reasonable clarifications to the items addressed in the original review. The responses and revisions to the manuscripts are acceptable, and I have no further revisions to add.

Reviewer #2 (Remarks to the Author):

This revised submission strengthens the presentation of the impressive body of work described by providing additional, helpful technical details. Assessment of efficacy remains a challenge given the modest experimental data – a single example – provided. More important, larger conceptual issues remain.

Fundamentally, we are interested in solving a registration problem. This requires our algorithms to identify correspondences between samples from the image datasets of interest. These samples may differ in scale (imaging resolution) and/or contrast (imaging physics) but also the biology imaged (imaging resolution and physics) – meaning it isn't just a matter of content becoming more apparent by increasing, say, the optical focus, but wholly different content that is revealed by, for example, a particular type of stain (even at the same resolution). It still isn't clear exactly how the latter aspects of biology revealed at different resolutions and/or contrast mechanisms are modeled via the proposed surjective transformations, which ultimately are only filtering operations. Given the lack of a biological basis or an encoding of biological knowledge in the proposed models (scattering transform, etc) in this work, the appropriate methodological context here seems to be the research in data-driven image translation and synthesis, for which deep learning currently dominates the field and with which comparison and contrast of this work is not made but would be most useful.

The application here to histological data notes tears and folds as problematic for registration. These features are challenging in part because the diffeomorphisms used to model the registration mapping assume the absence of tears and folds. Since the 'expanded' orbit of images allowed in this work presumably still cannot encode discontinuities (i.e., tears and folds), it isn't exactly obvious how improved performance is obtained.

As noted earlier, we are ultimately interested here in solving a registration problem. Given the active work in the field focused on the same application (histology to MRI registration), the authors might showcase what they believe to be the specific features of the current work that translate to improved performance over available approaches or address their known limitations.

Reviewer #3 (Remarks to the Author):

The authors have addressed the suggestions carefully and effectively. This manuscript described a non-rigid co-registration work between MRI and digital pathology which is a challenging task. The authors revising to the manuscript will definitively improve its readability and broaden the readership when the manuscript is published.

Dear Reviewers:

Please find point-by-point responses to each of your comments below. For convenience, we have copied over each of your comments in black. We indicate our responses with **red text**, and our alterations to the manuscript are indicated here in **blue text** with corresponding line numbers. These changes have been incorporated into the resubmitted manuscript draft in red text. Note that we have copied over the changes from our latex document, so equations and references are illustrated in that format. Also note that we provide full citations for additional references we've added to our manuscript in the blue text below, but references that were included previously are cited where applicable in the altered manuscript text by their short, in text, form only.

Thank you for your commitment to reviewing this manuscript. We believe the suggestions and questions you have posed have stimulated significant improvements.

Response to Reviewer 2 Comments

This revised submission strengthens the presentation of the impressive body of work described by providing additional, helpful technical details. Assessment of efficacy remains a challenge given the modest experimental data – a single example – provided. More important, larger conceptual issues remain.

Fundamentally, we are interested in solving a registration problem. This requires our algorithms to identify correspondences between samples from the image datasets of interest. These samples may differ in scale (imaging resolution) and/or contrast (imaging physics) but also the biology imaged (imaging resolution and physics) – meaning it isn't just a matter of content becoming more apparent by increasing, say, the optical focus, but wholly different content that is revealed by, for example, a particular type of stain (even at the same resolution). It still isn't clear exactly how the latter aspects of biology revealed at different resolutions and/or contrast mechanisms are modeled via the proposed surjective transformations, which ultimately are only filtering operations. Given the lack of a biological basis or an encoding of biological knowledge in the proposed models (scattering transform, etc) in this work, the appropriate methodological context here seems to be the research in data-driven image translation and synthesis, for which deep learning currently dominates the field and with which comparison and contrast of this work is not made but would be most useful.

We thank the reviewer for their assessment of the scattering transform and suggestion to comment on its comparison to other deep learning methods in the field. We agree with the reviewer that the scattering transform does not fundamentally encapsulate the biological connections between what is imaged at the micron scale in digital pathology (i.e. cells, molecules, misfolded proteins) and what is imaged at the millimeter scale in MRI (i.e. gray matter, white matter, CSF). Instead, it can be regarded in the context of empirical models such as those in deep learning that have shown predictive power across tasks in biology, physics, computer vision, and linguistics (He et al). Indeed, the scattering transform has been shown to be mathematically equivalent to a convolutional neural network (CNN) with a prescribed architecture (Bruna and Mallat, 2013). Furthermore, much work has been done to show not only the theoretical equivalence between such networks and the scattering transform, but the comparable performance of state-of-the-art deep learning methods in tasks such as estimating parameters of non-Gaussian fields (Cheng et al) and importantly for our work, in a range of image recognition tasks (Bruna and Mallat, 2013; Oyallon et al, 2015, Oyallon et al 2017).

The key benefit of using a scattering transform in lieu of traditional deep learning methods is that it is efficient to compute given the predefined choice of wavelets rather than the learning of all parameters, as is often done in the setting of deep learning. In other words, the scattering transform can build representations of data (images) analogous to those built implicitly in learning networks, but requires no training data nor extensive computational resources to do so, which we believe is particularly important in the setting of mapping histological contrast to MR contrast given the limited availability of data in digital pathology.

Here, as the reviewer correctly states, we are interested in solving a registration problem. While a biology-based model predicting one contrast from another may be desirable in the future to unravel the direct connections between qualities of anatomy at the micron versus millimeter scale, the difference in contrasts between modalities in this setting is treated as a nuisance variable that must be estimated to achieve accurate registration. Therefore, a data-driven approach such as those of deep learning is, again, as the reviewer notes, the appropriate set of methodologies to consider for this task. Others have shown success at estimating a mapping of histological contrast to MRI using random forest models (Iglesias et al) and deep learning networks (Yushkevich et al, 2021). Many such models require extensive computing time and resources not just to tune parameters, but to generate adequate training data for the learning task. Here, we advocate the use of the scattering transform for building correspondences between different contrast modalities given its demonstrated performance equal to that of deep learning methods in other image recognition tasks but with the benefits of requiring 1) fewer computational resources and time, 2) no additional training data, and 3) images not necessarily at the same resolution, as have been required by learning methods in previous approaches. This is particularly advantageous in the setting of estimating correspondences between high resolution histological contrast and lower resolution MRI contrast given the limited availability of readily available training data.

Finally, while the Scattering Transform does not aim to model the inherent physical/biological relationship between texture observed in images at the micron scale (e.g. cells and molecules) to that observed in images at the millimeter scale (e.g. gray and white matter), we choose to explicitly model the physics of the imaging technologies through our projection operator, here. In so doing, we build an overall method of registration between modalities that couples physics-based modeling with those of machine learning. Such coupling has been advocated by Karniadas et al in a range of inverse problems across biology and physics, as it greatly increases the efficiency of estimating solutions, which we believe is particularly important in cases, such as ours, where training data is limited.

We have added the following text and references to our Section 2.2 to elucidate the connection of the Scattering Transform to deep learning methods.

Lines 262-273: **Consequently, comparable representation of one stain to another and particularly, of any of these stained images to other modalities requires both crossing contrasts and scales of information. To achieve this, we harness Mallat's Scattering Transform to downsample histology images specifically nonlinearly, thereby capturing textural information at high resolution scales. The Scattering Transform has been shown to be equivalent in structure to CNNs with a particular architecture \cite{joanmallat2013}, but without additional training data or extensive time and resources needed to be computed. As a result, it has shown predictive power on par with state-of-the-art deep learning methods in a range of image recognition tasks \cite{oyallon2015deep,oyallon2017scaling}.**

Oyallon, E., Mallat, S.: Deep roto-translation scattering for object classification. In: Proceedings of the IEEE Conference on Computer Vision and Pattern Recognition, pp. 2865–2873 (2015).

Oyallon, E., Belilovsky, E., Zagoruyko, S.: Scaling the scattering transform: Deep hybrid networks. In: Proceedings of the IEEE International Conference on Computer Vision, pp. 5618–5627 (2017).

Lines 279-281: **The Scattering Transform mimics the architecture of CNNs by propagating sample images through a series of** alternating wavelet convolutions and nonlinear modulus operators across scales \cite{joanmallat2013} (see Supplementary Note S.3).

We have also added the following text and references to our discussion regarding our coupling of physics-based modeling and empirical modeling as means of furthering the efficiency and generalizability of our registration methods to instances particularly of limited training data.

Lines 589-613: **In contrast to methods exclusively of deep learning, which have gained popularity in excelling over the years in a range of image classification and processing tasks \cite{he2016deep}, the coupling of approaches used here to achieve these two features falls within the domain of what has been described as ``physics-based machine learning" \cite{karniadakis2021physics}. The key technology used for crossing scales in this work, without losing high resolution information, is the Scattering Transform. As highlighted in Section \ref{crossModal}, the Scattering Transform can be considered amongst the class of deep learning and other data-driven empirical models. In contrast, the projection operator we've introduced through Projective LDDMM aims to model explicitly the physics of the imaging technologies used. As such, we effectively couple empirical modeling with physics-based modeling, which has been shown to be more accurate and efficient than either type of approach on its own, particularly in settings of limited training data, as is frequently encountered across applications in biomedicine \cite{karniadakis2021physics}. ... Hence, in accommodating a range of imaging technologies and appealing to settings of limited data availability, we believe this model generalizes to many applications of image registration and integration across the sciences.**

He, K., Zhang, X., Ren, S., Sun, J.: Deep residual learning for image recognition. In: Proceedings of the IEEE Conference on Computer Vision and Pattern Recognition, pp. 770-778 (2016).

Karniadakis, G.E., Kevrekidis, I.G., Lu, L., Perdikaris, P., Wang, S., Yang, L.: Physics-informed machine learning. Nature Reviews Physics 3(6), 422–440 (2021)

The application here to histological data notes tears and folds as problematic for registration. These features are challenging in part because the diffeomorphisms used to model the registration mapping assume the absence of tears and folds. Since the 'expanded' orbit of images allowed in this work presumably still cannot encode discontinuities (i.e., tears and folds), it isn't exactly obvious how improved performance is obtained.

We thank the reviewer for their inquiry. The reviewer correctly notes that the expanded orbit of images generated through the scattering transform is not meant to resolve discontinuities such as tears and folds between different modalities. Here, as described above, the scattering transform is used to improve performance in estimating correspondences between histological and MR contrast without requisite training data and computational expense.

Discontinuities, as in our previous work, are treated through an EM framework by estimating the likelihood of each pixel being matching tissue and consequently lowering the relative weight of these discontinuities

in our cost function. As a result, discontinuities do not disrupt registration of nearby areas of in tact tissue but are not modeled explicitly, as the reviewer notes, with our diffeomorphisms. Modeling the actual discontinuities in tissue explicitly would be an extension of this work that we believe would benefit from active research in the field of computational anatomy looking at mappings other than diffeomorphisms for relating objects of different topology.

We have added the following to our discussion to clarify our treatment of discontinuities and pose this possible extension to explicit modeling of them:

Lines 621-626: The mixtures are built to expand **upon** the orbit of images generated by the Scattering Transform by including non-foreground tissue as manifest by the smooth deformation of the MR template. **Effectively, they enable the estimation of smooth deformations between template and target with priority given to areas predicted to exhibit foreground tissue without artifacts. Explicit modeling of these artifacts and discontinuities through transformations other than diffeomorphisms and with treatment of the discordant topologies introduced between template and target images represents an area for future work.**

As noted earlier, we are ultimately interested here in solving a registration problem. Given the active work in the field focused on the same application (histology to MRI registration), the authors might showcase what they believe to be the specific features of the current work that translate to improved performance over available approaches or address their known limitations.

We thank the reviewer for their invitation to comment further on how the method(s) we present compare in performance to other current methods for registering histology images to MRI. First, as a mathematical modeling framework, we believe Projective LDDMM fills a gap in literature of a model of dense to sparse image captures that is general enough to encapsulate not just the relation of histological images to MRI but the relationships of image captures both within and across diverse imaging technologies such as PET, CT, optical sectioning, and light sheet microscopy, as described in Section 2.3 and Supplementary Note S.1. Second, this work introduces a scattering transform to cross modalities. This departs from previous work in simultaneously estimating contrast correspondences between images at different scales (Iglesias et al, Tward et al) and without the need of training data (Yushkevich et al). Through the nonlinear downsampling that occurs as part of the (Subsampled) Scattering Transform, key textural information encoded at high resolution histological scales is retained, which offers robustness in modeling relationships between more diverse and complex color profiles (e.g. different histological stains) of images at different scales. Additionally, the Scattering Transform is more efficient to compute with regard to computational time and resources compared with deep learning methods.

As noted above, a limitation this approach shares with others (Iglesias et al) is in the lack of explicit modeling of discontinuities in tissue. We estimate a lower weight for these locations in the matching term of our cost function which enables us to compute correspondences between histology and MRI without grave disruptions; however, direct modeling of these discontinuities might enable even more accurate correspondences to be estimated in the future.

We have made the following changes in our introduction (with added references) to emphasize the advantages of our approach through the use of the Scattering Transform and clarify its differentiation from our treatment of discontinuities in tissue.

Lines 090-114: **The second major contribution of this work is the introduction of a novel photometric transformation between modalities via a Scattering Transform that simultaneously**

achieves correspondence between contrasts at different scales, analogous to those captured by convolutional neural networks (CNNs), but without the need of training data or expensive computational resources \cite{joanmallat2013}. For alignment specifically of modes of histology to MRI, cross-modality similarity modelling is essential. Several strategies for representing image similarity have emerged including cross-correlation \cite{Avants}, mutual information \cite{Pluim2003}, and local textural characteristics \cite{Heinrich2012}. We previously used a polynomial transformation to accommodate crossing modalities at a single (low) resolution \cite{tward2020}. **Others have used a variety of machine learning approaches to cross contrasts both in 2D and 3D, again between images at the same scale \cite{iglesias2018joint,yang2020mri,islam2021deep}. Here, to accommodate crossing contrast modalities at different scales, particularly in a context of limited training data,** we expand histology image space to a span of **non-linear** discriminative filtered images via Mallat's Scattering Transform \cite{mallat2012,joanmallat2013}. These filtered images **are efficiently computed directly from the sample histology images. They are at the lower resolution of MRI, but represent local, and specifically non-linear,** radiomic textures at histological scales, **unlike typical linearly downsampled images,** and thus, can effectively be used to predict MRI contrast.

As histological images carry large numbers of imperfections with tears, image stitching, and lighting variations, **we additionally estimate high-dimensional geometric transformations in the image plane through the introduction of Gaussian mixture models. As in previous work \cite{tward2020}, these Gaussian mixtures models interpret image locations in each histological slice as matching tissue, background, or artifact. Here, we depart from this and other previous work \cite{Lee2018,tward2020} in estimating high-dimensional diffeomorphisms in the image plane rather than rigid motions to encapsulate better the distortion that occurs following tissue sectioning in histological processing.** We proceed by way of the Expectation-Maximization (EM) algorithm \cite{dempster1977} in estimating deformations that prioritize image matching at locations that are, in turn, estimated more likely to be matching tissue.

Iglesias, J.E., Modat, M., Peter, L., Stevens, A., Annunziata, R., Vercauteren, T., Lein, E., Fischl, B., Ourselin, S., Initiative, A.D.N., et al.: Joint registration and synthesis using a probabilistic model for alignment of mri and histological sections. *Medical image analysis* 50, 127–144 (2018).

Yang, Q., Li, N., Zhao, Z., Fan, X., Chang, E.I., Xu, Y., et al.: Mri cross-modality image-to-image translation. *Scientific reports* 10 (1), 1–18 (2020).

Islam, K.T., Wijewickrema, S., O'Leary, S.: A deep learning based framework for the registration of three dimensional multi-modal medical images of the head. *Scientific Reports* 11(1), 1–13 (2021).

We have also added the references and text noted in our response to the reviewer's first comment to the discussion to emphasize what we believe to be the key advantages, here, of our methods over previous work.

Finally, we have added an additional figure (Figure 3) that highlights the added information encoded in the Scattering Transform dimensions compared with linear downsampling that results in a single average signal. Associated with this figure, we've added the following explanatory text and figure caption:

Lines 316-327: Through its nonlinear downsampling, the (subsampled) Scattering Transform has the power to distinguish compartments from one another that appear with single contrast in the linearly downsampled image. Examples of this discrimination are shown in Figure \ref{scatFigure} both for Nissl and PHF-1 stained histology images. In the average Nissl image, for instance, the white matter alveus

has similar contrast to the neighboring hippocampus and background space, preventing these separate structures from being mapped to different corresponding contrasts in other modalities, such as MRI. Likewise, the separate regions in the hippocampal formation (dentate gyrus and CA fields) appear with similar contrast in the average PHF-1 image, masking the boundary between them, whereas the scattering image clearly manifests this boundary.

Figure Caption: (Left) Average signal resulting from linear downsampling of Nissl (top) and PHF-1 (bottom) stained histology images at 0.002 mm resolution to 0.067 mm resolution. (Middle) Scattering images at 0.067 mm resolution projected onto 3 PCA dimensions and mapped to RGB. Specific compartments highlighted where scattering representation shows distinguishable boundaries not in average signal. (Right) High-field MRI slice and manual segmentations showing corresponding contrast and subregion delineations of anatomy in first two columns.

REVIEWERS' COMMENTS:

Reviewer #2 (Remarks to the Author):

The provided response adequately addresses all of the issues raised in prior critique. The corresponding revision of the manuscript adds significant clarity to the presentation, including the novel elements of the work.